# Scorpion Venom as a Source of Cancer Drugs: A Comprehensive Proteomic Analysis and Therapeutic Potential

**DOI:** 10.3390/ijms26209907

**Published:** 2025-10-11

**Authors:** Stephanie Santos Suehiro Arcos, Mariana Ramos da Cunha Aguiar, Júlia de Oliveira, Matheus Ramos da Silva, Isabela de Oliveira Cavalcante Pimentel, Nicolas Gamboa dos Anjos, Gustavo Henrique Rohr Souza Machado, Kimberly Borges Evangelista, Fernanda Calheta Vieira Portaro, Leo Kei Iwai

**Affiliations:** 1Laboratory of Applied Toxinology (LETA) and Center of Toxins, Immune-Response and Cell Signaling (CeTICS), Butantan Institute, São Paulo 05503-900, Brazil; 2Laboratory of Structure and Functions of Biomolecules, Butantan Institute, São Paulo 05503-900, Brazil; fernanda.portaro@butantan.gov.br

**Keywords:** proteomics, cancer, scorpion venom, natural products, anticancer peptides, targeted therapy, drug delivery

## Abstract

Scorpion venom is a rich source of bioactive compounds with significant potential for anticancer drug development. Its diverse molecular composition, including neurotoxins, antimicrobial peptides, and enzymes, provides a vast library for therapeutic innovation. Proteomic analyses have characterized venom composition in several species, while further functional assays have clarified their anticancer mechanisms. This review synthesizes current knowledge on scorpion venom-derived peptides with demonstrated anticancer activity, which selectively target ion channels, induce apoptosis, or disrupt tumor microenvironments. Where available, we highlight proteomic studies that have identified these components and discuss their structural features relevant to drug design. We also examine clinical applications and the challenges in translating venom peptides into therapies. The crucial and growing role of proteomics in this field, particularly for venom fractionation, component identification, and structural characterization, is critically evaluated.

## 1. Introduction

### 1.1. The Global Cancer Burden

Cancer remains a leading cause of mortality worldwide, with 20 million new cases and 9.7 million deaths reported in 2022 [1]. Projections suggest these numbers will increase significantly by 2040, with an estimated 29.9 million new cases annually, driven by population growth, aging, and changing exposure patterns to risk factors [1]. Despite significant advances in cancer treatment, conventional therapies present major limitations, including systemic toxicity, drug resistance, and restricted delivery across the blood–brain barrier, and high costs, which negatively impact patient quality of life and access to care, particularly in low- and middle-income countries [2,3,4]. Cancer is inherently complex, representing a group of related yet distinct diseases, each with unique molecular and cellular features [4]. Conventional chemotherapeutic agents, while effective against rapidly dividing cells, often damage healthy tissues, leading to adverse effects such as myelosuppression, gastrointestinal toxicity, cardiotoxicity, neurotoxicity, secondary malignancies, and fertility problems [4,5,6]. Tumor resistance mechanisms, including enhanced drug efflux, alterations in drug targets, increased DNA repair, activation of alternative survival pathways, metabolic reprogramming, and modifications in the tumor microenvironment, further limit therapeutic efficacy [4,6,7,8,9]. Physical barriers such as the blood–brain barrier pose additional challenges for treating central nervous system (CNS) tumors [3,10].

Given these challenges, peptides derived from scorpion venom have attracted attention as potential anticancer agents. These compounds exhibit high affinity for ion channels and receptors dysregulated in tumor cells, together with selective cytotoxicity. This review presents an analysis of the proteomics of scorpion venoms, their multimodal anticancer mechanisms, and their potential as therapeutic agents in oncology.

### 1.2. Natural Products and Venom-Drived Compounds

Natural products have historically been a major source of drug discovery, with approximately 32.5% of drugs approved between 1981 and 2019 either derived from or inspired by natural sources [11]. Plant-derived compounds, such as taxanes and vinca alkaloids, have transformed oncology, while animal venoms remain relatively underexplored. Venoms provide evolutionarily optimized peptides that target ion channels and receptors altered in cancer [12], exhibiting structural complexity, diverse mechanisms of action, and favorable therapeutic indices [13,14]. The growing interest in venom-derived peptides, particularly from scorpions, follows logically from these precedents, as they offer biologically refined molecules capable of addressing the shortcomings of conventional therapies.

### 1.3. Historical Use of Venom in Medicine

Animal venoms have been utilized in traditional medicine for millennia, treating pain, inflammation, neurological disorders, cardiovascular diseases, and cancer [12,13]. Several venom-derived drugs are clinically approved: captopril, from the jararaca viper (*Bothrops jararaca*) venom [15,16]; eptifibatide (Integrilin^®^), from the pygmy rattlesnake (*Sistrurus miliarius barbourin*) venom [17]; ziconotide (Prialt^®^), from the marine mollusk (*Conus magus*) venom [18]; and exenatide (Byetta^®^), from the Gila monster (*Heloderma suspectum*) venom [19]. These examples illustrate the potential of venom-based therapeutics and provide a rationale for investigating scorpion toxins as targeted anticancer agents.

## 2. Literature Search Strategy and Scope of the Review

To provide a comprehensive analysis of the role of scorpion venom in oncology, with a particular focus on the contributions of proteomics, this article presents a narrative review of the current scientific landscape. The information synthesized in the following sections was gathered through a literature search of the PubMed and Web of Science databases to identify studies on the composition, bioactive peptides, molecular mechanisms, immunomodulatory effects, diagnostic applications, drug development, and clinical evaluation of scorpion venom. Keywords included “scorpion venom”, “peptides”, “anticancer”, “proteomics”, “venomics”, “mechanism”, “immunomodulation”, “diagnostic”, “drug development”, and “clinical trial”.

The selection process prioritized original preclinical and clinical studies detailing the structural, biochemical, or proteomic characterization of venom components and their therapeutic or diagnostic applications. To maintain a focus on primary data and robust findings, general reviews, editorials, conference abstracts, and studies lacking sufficient methodological detail were excluded. After duplicate removal, articles were screened based on title, abstract, and full text. Extracted data comprised study type, venom source, experimental model, bioactive compounds, molecular targets, and observed effects. Additional resources, UniProt and PubChem, were consulted to retrieve proteomic and chemical information on venom-derived compounds. As this manuscript is intended as a narrative review offering an expert overview, its methodology differs from that of a systematic review, and therefore a PRISMA protocol was not utilized.

The following sections will now delve into the composition of scorpion venom and its most promising anticancer peptides, the discovery and characterization of which have been largely driven by proteomic advancements.

## 3. Composition and Biochemistry of Scorpion Venom

Scorpion venom represents a rich reservoir of bioactive compounds, comprising diverse proteins, peptides, enzymes, nucleotides, lipids, inorganic salts, free amino acids, and small molecules, demonstrating remarkable specificity in targeting cellular pathways [20,21]. Among these, proteins and peptides are the most pharmacologically significant, particularly for anticancer applications due to their high target selectivity and low off-target toxicity [22,23].

### 3.1. Proteomic Characterization of Scorpion Venom

Recent advances in high-throughput proteomics have revolutionized the comprehensive molecular characterization of scorpion venom, a field often referred to as venomics, enabling profiling of its complex molecular composition. State-of-the-art analytical strategies, such as nano-scale liquid chromatography coupled with tandem mass spectrometry (LC-MS/MS), combined with advanced bioinformatics pipelines, have enabled researchers for the in-depth profiling of this complex natural cocktail, revealing hundreds of unique proteins and peptides that form a vast library of bioactive compounds [24,25] (Table 1).

The standard venomics workflow, as summarized in Figure 1, is a multi-step process. It begins with the fractionation of crude venom using chromatographic techniques, such as high-performance liquid chromatography (HPLC) or size exclusion chromatography (SEC) to reduce the sample’s complexity. These fractions are then analyzed using high-resolution mass spectrometry, most commonly nano-scale liquid chromatography coupled with tandem mass spectrometry (LC-MS/MS). This core step allows for the precise sequencing of peptides within the venom [24,25]. The resulting data are then matched against specialized protein databases such as UniProt and VenomZone, and interpreted through advanced bioinformatics pipelines. This systematic approach is crucial for identifying known toxins and discovery novel bioactive molecules with therapeutic potential, including those with anticancer properties [81].

Beyond simple identification, proteomic analyses provide deep insights into the diversity and evolution of venom composition across different scorpion species, highlighting conserved protein families that are promising leads for drug development [82,83]. Furthermore, quantitative proteomic methods such as label-free, isobaric tags for relative and absolute-quantification (iTRAQ), and tandem mass tag (TMT)-based approaches, reveal the relative abundance of different venom constituents, offering clues about their variability across species, environmental conditions, and biological importance [84].

These techniques are also essential for identifying post-translational modifications (PTMs), such as disulfide bonds, phosphorylation, and N-glycosylation. PTMs are critical for the structural stability and biological activity of venom toxins and are often missed by other analytical methods [85,86,87]. The integration of “top-down” (intact protein analysis) and “bottom-up/shotgun” (digested peptide analysis) proteomic strategies researchers can build a complete picture of venom components, providing complementary insights into venom protein structure and function, facilitating the identification of promising therapeutic candidates. Together, these proteomic advances not only deepen our fundamental understanding of venom biology but also accelerate the rational discovery of new antitumoral drug candidates.

### 3.2. Major Protein and Peptide Components of Scorpion Venom

The protein and peptide components of scorpion venom represent a complex mixture of bioactive molecules that constitute the molecular basis for its therapeutic potential. These biomolecules can be broadly classified into major groups based on their distinct structural and functional characteristics.

#### 3.2.1. Neurotoxins

Neurotoxins are among the most abundant and extensively characterized components of scorpion venom. They include long-chain toxins that modulate voltage-gated sodium channels and short-chain toxins that target potassium and chloride channels [23,88,89,90,91,92,93,94,95,96,97,98].

The therapeutic potential in oncology is linked to the modulation of ion channels that are often dysregulated in cancer cells, disrupting membrane potential and inducing apoptosis [99,100]. Some short-chain neurotoxins can also cross the blood–brain barrier and selectively bind to cancer cell membranes, making them promising candidates for treating neurological malignancies [21,101,102,103].

#### 3.2.2. Antimicrobial and Cytolytic Peptides

Scorpion venom also contains antimicrobial peptides that, beyond their antimicrobial effects [104], exhibit potential anticancer properties through membrane-disruptive mechanisms [73,105,106,107]. These peptides often exhibit selective cytotoxicity against cancer cells while showing minimal effects on normal cells, a property that significantly enhances their therapeutic relevance [75].

#### 3.2.3. Enzyme Inhibitors

Enzyme inhibitors represent another bioactive class in scorpion venom, particularly those targeting proteases that play crucial roles in tumor growth, invasion, and metastasis [76,108,109,110,111,112,113]. By modulating matrix metalloproteinases (MMPs) and serine proteases involved in extracellular matrix remodeling, these inhibitors can suppress cancer cell migration and angiogenesis, highlighting their potential as adjuvants in anticancer therapy.

#### 3.2.4. Disulfide-Rich Peptides

The disulfide-rich peptides (DRPs) are of particular pharmacological interest due to their remarkable structural stability, conferred by multiple disulfide bonds, and their high specificity for molecular targets, features that make them attractive drug leads [85,114,115,116].

Recent discoveries reveal peptides capable of modulating tumor suppressor *p53* expression or interfering with the PI3K/Akt pathway, which is frequently dysregulated in cancer [22,117,118,119]. Additionally, the venom contains various bioactive amines and polyamines that can also modulate cellular signaling and act synergistically with other venom components to enhance cytotoxic effects [120,121,122,123,124].

#### 3.2.5. Enzymatic Components

The enzymatic components of scorpion venom comprise a diverse set of catalysts that play a pivotal role in its biological activity and therapeutic potential [84]. Among these, phospholipases, particularly phospholipase A2 (PLA_2_), are key mediators of membrane remodeling and initiators of cellular responses associated with anticancer effects s [100,125,126,127,128,129,130,131]. Hyaluronidases, often referred to as “spreading factors,” degrade hyaluronic acid within the extracellular matrix (ECM), facilitating the diffusion of other venom components and enhancing their bioavailability, a property that can be harnessed to improve targeted drug delivery [132]. Metalloproteases modulate the ECM, thereby affecting cell adhesion and migration, processes that are critical in cancer invasion and metastasis [132,133,134]. Serine proteases add further complexity to the venom’s enzymatic profile, by participating in coagulation cascades and modulating inflammatory pathways [135].

The coordinated action of these enzyme systems creates a complex network of biochemical interactions with significant therapeutic applications. A comprehensive understanding of their catalytic properties, regulatory mechanisms, and biological targets is crucial for the rational development of novel anticancer strategies based on scorpion venom components [136].

## 4. Molecular Mechanisms of Anticancer Activity

Scorpion venom components exert anticancer effects through diverse molecular mechanisms, including induction of apoptosis, cell cycle regulation, and modulation of ion channels. Rather than acting as non-specific cytotoxins, many peptides demonstrate selectivity for malignant cells by targeting signaling pathways, surface receptors, or membrane properties that are dysregulated in cancer. The following subsections provide an overview of the principal mechanisms, illustrated by representative venom-derived molecules (Figure 2). Detailed descriptions of individual peptides are provided in Section 5.

### 4.1. Induction of Apoptosis

The investigation of cell death pathways induced by scorpion venom components has revealed complex molecular mechanisms of action that selectively induce cancer cell death through both intrinsic and extrinsic apoptotic pathways. The intrinsic apoptotic pathway activation involves critical disruption of mitochondrial function, as demonstrated by Smp24, a peptide derived from *Scorpio maurus palmatus*. This peptide disrupts mitochondrial homeostasis by reducing the mitochondrial membrane potential (ΔΨm), increasing reactive oxygen species (ROS) production, and inducing cytoskeletal reorganization in HepG2 cancer cells while exhibiting minimal toxicity toward normal LO2 liver cells [71,75]. Mitochondrial depolarization and ROS generation are hallmarks of caspase-3-mediated disruption of electron transport chain complexes I and II, a feedback mechanism that amplifies apoptotic signaling after cytochrome c release [26,137]. In addition, both Smp24 and Smp43 activate caspase-1, triggering pyroptosis in both myeloid (KG1-a) and lymphoid (CCRF-CEM) leukemia cells, again with minimal toxicity to non-tumor HaCaT cells [73].

Bengalin from *Heterometrus bengalensis* Koch exemplifies mitochondrial-targeting peptides, inducing cytochrome c release and caspase activation in leukemic cells [35]. Detailed proteomic and structural insights are discussed in Section 5.1.2. A similar mechanism is seen with peptide BmKn-2 from *Mesobunthus martensii* Karsch, which induces apoptosis in canine mammary gland tumor CHMp-5b and CHMPp-13a cell lines and in both human oral squamous carcinoma cells (HSC4) and human mouth epidermoid carcinoma cells (KB) via Bax/Bcl-2 modulation and caspase 9 activation, while sparing normal human gingival and dental pulp cells [44,45]. Consistently, *Hemiscorpius lepturus* venom promotes Bax, caspase 3, and p53 overexpression alongside Bcl2 suppression in both CT26 colon carcinoma cells and xenograft tumors, with low cytotoxicity toward non-tumorigenic VERO cells [54].

Venoms also act on oncogenic signaling. *Buthus martensii* Karsch venom (BmK) selectively kills Raji and Jurkat lymphoma cells while sparing normal human peripheral blood lymphocytes. In Raji cells, BmK upregulated the tumor suppressor phosphatase and tensin homolog (PTEN) expression, decreasing Akt and Bad phosphorylation, thereby downregulating PI3K/Akt signaling. In PTEN-negative Jurkat cells, apoptosis proceeds through p27-mediated mechanisms, underscoring cell-specific vulnerabilities. Combining BmK with the Akt inhibitor LY294002 synergistically enhanced apoptosis, underscoring the therapeutic potential of venom-derived compounds in targeting oncogenic pathways [42].

The extrinsic apoptotic pathway is mediated through interactions with death receptors on cancer cell surfaces. Neopladines 1 and 2 from *Tityus discrepans* venom upregulate Fas ligand in SKBR3 breast cancer cells, promoting death-inducing signaling complex (DISC) formation, caspase-8 activation, and selective killing relative to non-malignant MA104 cells [66]. Similar receptor-driven effects have been reported in *Androctonus crassicauda* venom-treated HCT-8 colorectal cancer cells [22,26]. Activated caspase-8 also cleaves the BH3-only protein Bid, generating truncated Bid (tBid) that translocates to mitochondria, linking extrinsic and intrinsic pathways, thereby amplifying mitochondrial permeabilization and caspase-3 activation [22,26,138,139,140,141,142].

Taken together, these findings show that scorpion venoms exploit multiple cell death pathways, with selectivity for cancer cells driven by mitochondrial dysfunction, death receptor activation, and pathway cross-talk. Their multi-target nature, simultaneously inhibiting proliferation, angiogenesis, and metastasis, positions them as promising candidates for combinatorial anticancer therapies [43,143].

### 4.2. Cellular Signaling and Cycle Disruption

The impact of scorpion venom on cellular signal transduction pathways has been mapped through comprehensive phosphoproteomic analyses, revealing intricate networks of molecular interactions. Key targets include MAP kinases, PI3K/Akt, and JAK/STAT pathways, with venom peptides selectively inhibiting growth factor receptor signaling, disrupting crucial survival pathways in cancer cells. For example, *Buthus martensii* Karsch venom (BmK) upregulates PTEN in Raji lymphoma cells, suppressing PI3K/Akt via reduced Akt/Bad phosphorylation, while p27 mediates death in PTEN-negative Jurkat cells [42]. Temporal proteomics reveals sequential pathway activation, while protein–protein interaction studies highlight venom-induced disruption of cross-talk between survival pathways, potentially circumventing drug resistance.

Proteomics also clarifies cell cycle interference, identifying venom interactions with cyclins, cyclin-dependent kinases (CDKs), and checkpoint regulators. High-throughput screens show venom peptides arresting the cell cycle at G1/S or G2/M transitions through post-translational modifications, such as phosphorylation of regulatory proteins. Quantitative proteomics demonstrates differential expression of cycle-related proteins in cancer versus normal cells, supporting selective toxicity. These insights have propelled targeted strategies to halt cancer proliferation.

## 5. Scorpion Venom Peptides: Anticancer Activities and Mechanisms of Tumor Targeting

### 5.1. Apoptosis Induction

#### 5.1.1. TsAP-1 and TsAP-2 from *Tityus serrulatus*

The 17-mer peptides TsAP-1 and TsAP-2, isolated from the venom of the Brazilian yellow scorpion *Tityus serrulatus*, demonstrate significant anticancer potential through selective cytotoxicity and apoptosis induction (see Section 4.1). While TsAP-1 exhibits modest activity against oral carcinoma (H157 cells, IC_50_ > 50 μM), its cationic analog TsAP-S1 shows dramatically enhanced potency (IC_50_ = 2.5 μM in leukemia cells) through membrane disruption mechanisms [78]. TsAP-2 displays greater inherent activity, with an IC_50_ of 4 μM in SKBR3 breast cancer cells, and its engineered analog TsAP-S2 achieves remarkable potency (IC_50_ = 0.83 μM) while maintaining selectivity for cancer cells over normal cells. These peptides induce apoptosis through mitochondrial disruption and activation of caspase pathways, as evidenced by their ability to suppress proliferation in multiple cancer cell lines, including glioblastoma (U251-MG) and prostate adenocarcinoma (PC-3), with minimal effects on non-malignant cells [78]. Further supporting their therapeutic potential, TsAP-2 and the structurally related peptide Stigmurin from *Tityus stigmurus* exhibit antiproliferative effects on tumor cells while demonstrating low cytotoxicity toward normal cells, highlighting their selectivity and safety profile [77]. The enhanced activity of cationicity-modified analogs (TsAP-S1 and TsAP-S2) underscores the importance of structural optimization for improving anticancer efficacy. These findings position TsAP-1 and TsAP-2 as promising candidates for targeted cancer therapy, with their mechanisms of action rooted in membrane interaction and apoptotic pathway activation. Further research is needed to elucidate their precise molecular targets and evaluate their efficacy in vivo.

#### 5.1.2. Bengalin from *Heterometrus bengalensis* and Other Novel Peptides

Bengalin, a peptide isolated from the Indian black scorpion *Heterometrus bengalensis*, exemplifies how structural and proteomic analyses deepen our understanding of venom-derived anticancer compounds. Proteomic profiling has confirmed its abundance in crude venom and provided evidence of its selective enrichment in venom fractions with cytotoxic activity [144]. Structurally, Bengalin exhibits a disulfide-stabilized fold characteristic of scorpion toxins, conferring conformational stability and resistance to proteolytic degradation, properties that enhance its therapeutic potential [33]. Beyond its structural resilience, proteomic and immunoblotting studies have documented Bengalin’s dual impact on cancer cells. It induces apoptotic signaling, evidenced by Bax/Bcl-2 modulation, cytochrome c release, and caspase cascade activation, while simultaneously engaging autophagic responses through Beclin-1 and Atg upregulation and LC3 conversion [34,35].

These molecular features correlate with selective cytotoxicity by inducing apoptosis in U937 and K562 leukemic cells through multiple pathways mediated by the increase in caspase-3 activity and mitochondrial disruption pathways [35,36] and autophagic cell death via inhibition of proliferative MAPK/ERK and PI3K/AKT pathways [34,35]. Although Bengalin has shown selective cytotoxic potential for leukemic cells [144], more studies are needed to investigate the exact mechanisms of action of this compound and perform in vivo assays to confirm its anti-tumorigenic activity.

#### 5.1.3. Neopladine 1 and Neopladine 2 from *Tityus discrepans*

Neopladine 1 and 2, isolated from *Tityus discrepans* venom, represent novel anticancer compounds with unique mechanisms of action. These peptides exhibit selective anticancer activity against HER2-positive human breast carcinoma SKBR3 cells, inducing apoptosis while showing negligible effects on non-malignant MA104 monkey kidney cells [37,66,122]. Mass spectrometry analysis determined their molecular masses to be 29,918 Da (Neopladine 1) and 30,388 Da (Neopladine 2), and their N-terminal sequences were confirmed by Edman degradation. The peptides induce apoptosis in SKBR3 cells, with Neopladine 1 causing 6.3% apoptosis and Neopladine 2 causing 4.1% apoptosis after 5 h of exposure; prolonged exposure increases this effect. Immunohistochemical studies indicate that neopladines bind to SKBR3 cell surfaces, upregulating Fas Ligand (FasL) and Bcl-2 expression, which are critical in apoptosis signaling. Interestingly enough the combined application of neopladines 1 and 2 reduced apoptosis but increased necrosis, suggesting complex interactions that requires additional investigation [66].

### 5.2. Ion Channel Modulation

#### 5.2.1. AGAP-SYPU2 from *Buthus martensii* Karsch

Analgesic Antitumor Peptide AGAP-SYPU2, a peptide isolated from *Buthus martensii* Karsch (BmK) scorpion venom, exhibits dual analgesic and antitumor activities [29,145]. Shao et al. purified AGAP-SYPU2 and demonstrated its strong analgesic effects against both visceral and somatic pain, with its mechanism involving inhibition of voltage-gated sodium channels, which are critical in pain signaling. Although its onset of action is slower than morphine, it provides prolonged analgesic effects [30].

In cancer therapy, AGAP (a homolog of AGAP-SYPU2) shows promising antitumor properties. It prolonged survival by 36.05% in Ehrlich ascites tumor models and reduced tumor weight by 46.3% in S180 fibrosarcoma mouse models [30]. The peptide’s anticancer activity is linked to its modulation of sodium channels such as Nav1.4, Nav1.5, and Nav1.7, which are overexpressed in cancer cells and contribute to proliferation and migration. Moreover, AGAP was able to inhibit the proliferation and migration of SHG-44 glioma cells, suggesting a shared mechanism involving sodium channel blockade [31].

Although the direct receptor target of BmK AGAP has not been identified, evidence indicates it downregulates pentraxin-3 (PTX3), a key mediator of receptor–ligand interactions in the tumor microenvironment. This downregulation suppresses the NF-κB/Wnt/β-catenin axis, ultimately reducing stemness and epithelial–mesenchymal transition (EMT) in breast cancer cells [145].

The dual functionality of AGAP-SYPU2, both as an analgesic and antitumor agent, makes it clinically valuable, as it may improve patient survival without compromising quality of life. However, further research is needed to optimize its potency and evaluate its efficacy in other cancer types.

#### 5.2.2. BotCl from *Buthus occitanus tunetanus*

BotCl, a chlorotoxin-like peptide isolated from the venom of the scorpion *Buthus occitanus tunetanus*, has emerged as a promising anticancer agent due to its ability to target ClC-3 chloride channels, which are overexpressed in glioblastoma and breast cancer cells [46,48]. This peptide belongs to the chlorotoxin family, characterized by four disulfide bonds that confer structural stability and high binding affinity to tumor-specific ion channels and membrane receptors, such as matrix metalloproteinase-2 (MMP-2) [46,47]. BotCl shares significant sequence homology with chlorotoxin (CTX), a well-studied scorpion venom peptide currently in clinical trials for glioma imaging and therapy, suggesting similar mechanisms of action, including inhibition of tumor cell migration and invasion [46,47].

The anticancer properties of BotCl extend beyond glioblastoma. Its ability to reduce tumor viability in breast cancer models highlights its broad-spectrum potential, possibly through mechanisms involving chloride channel blockade and disruption of tumor microenvironment signaling [46,48]. Furthermore, BotCl’s structural stability under varying pH and temperature conditions enhances its suitability for therapeutic development, including conjugation with imaging agents or cytotoxic drugs for targeted cancer therapy [48,146].

In summary, BotCl represents a novel chlorotoxin-like peptide with significant anticancer potential, supported by proteomic and structural analyses. Its ability to target ClC-3 channels and MMP-2 in glioblastoma and breast cancer positions it as a promising candidate for further preclinical and clinical evaluation, particularly in the development of tumor-specific diagnostic and therapeutic agents.

#### 5.2.3. Iberiotoxin (IbTX) from *Hottentotta tamulus*

Iberiotoxin (IbTX), a 37-amino acid peptide derived from the venom of the scorpion *Hottentotta tamulus* (also known as *Buthus tamulus* or *Mesobuthus tamulus*), has emerged as a potent inhibitor of voltage-gated potassium channels, particularly Kv1.1 and Kv1.3, which are overexpressed in malignancies such as glioblastoma (U87), breast (MDA-MB-231), colon (LS174), cervical (HeLa), and ovarian (A2780) cancers [56,57,58]. This toxin shares 68% sequence homology with charybdotoxin (ChTX) but exhibits superior selectivity for large-conductance calcium-activated potassium (BKCa) channels, with an equilibrium dissociation constant (Kd) of 1.16 nM, making it one of the most potent blockers of this channel family [56,59,60,147].

IbTX acts on Kv1.1 and Kv1.3 potassium channels, causing calcium dysregulation and inducing apoptosis in cervical and ovarian cancers [148]. Beyond this direct channel blockade, IbTX-sensitive currents have been shown to modulate the proliferation of breast cancer cells (MCF-7, MDA-MB-231) under conditions of elevated intracellular calcium, such as during ATP stimulation [57]. This suggests a context-dependent role in tumor signaling, where IbTX could synergize with calcium-mobilizing therapies. Additionally, structural studies using synthetic chimeric peptides, such as IbTX-ChTX hybrids revealed that the toxin’s C-terminal domain is critical for Kv1.1/1.3 selectivity, providing a template for designing targeted anticancer derivatives [148].

Despite its promise, challenges remain in translating IbTX into clinical applications due to its potential off-target effects on neuronal and cardiovascular Kv channels. However, advancements in bioconjugation, such as biotinylated IbTX for imaging BKCa distribution in cancer cells, highlight its utility as a molecular tool for tumor profiling and drug development [149]. Future research should explore engineered analogs with enhanced tumor specificity and delivery systems to harness IbTX’s full therapeutic potential.

In summary, IbTX represents a structurally and functionally characterized scorpion venom peptide with validated anticancer activity, supported by proteomic and electrophysiological evidence. Its ability to target oncogenic potassium channels positions it as a promising candidate for further preclinical evaluation in calcium-driven malignancies [56,57,59].

### 5.3. Cell Cycle Arrest

#### 5.3.1. Gonearrestide from *Androctonus mauritanicus*

Gonearrestide, an 18-amino acid peptide (2.2 kDa) isolated from the venom of *Androctonus mauritanicus*, represents a novel class of scorpion venom-derived compounds with potent anticancer activity. Identified through a high-throughput platform combining next-generation sequencing (NGS) transcriptomics and LC-MS/MS proteomics, this peptide was selected from 238 novel peptides discovered in scorpion venom libraries due to its selective cytotoxicity against cancer cells while sparing normal epithelial cells and erythrocytes [53].

The peptide’s mechanism of action centers on inducing G1 cell cycle arrest in colorectal cancer cells (HCT116), achieved through dual modulation of cyclin-dependent kinase (CDK) regulators. Gonearrestide downregulates CDK4, a key driver of G1/S transition, while simultaneously upregulating the CDK inhibitors p21 and p27, as well as cyclin D3 [53]. This coordinated action effectively halts cancer cell proliferation, as demonstrated by RNA sequencing showing significant alterations in cell cycle-related gene expression profiles.

In preclinical validation, Gonearrestide exhibited broad-spectrum activity against multiple cancer cell lines while showing negligible toxicity to normal cells, a critical advantage over conventional chemotherapeutics. Its ability to inhibit primary colon cancer cells and solid tumors in vivo further underscores its therapeutic potential. The discovery of Gonearrestide exemplifies how integrated omics technologies (proteomics/transcriptomics) can accelerate the identification of bioactive venom peptides with precise mechanisms of action. Its cell cycle-specific targeting offers a template for developing novel anticancer agents that combine high potency with reduced off-target effects.

#### 5.3.2. PESV from *Buthus martensii* Karsch

Mass spectrometry and molecular biology analyses of the *Buthus martensii* Karsch (BmK) venom have identified multiple bioactive components, including polypeptides with anticancer activity [115,150]. Comprising 50–60 amino acid peptides, the Polypeptide Extract from Scorpion Venom (PESV) interferes with tumor growth through both antiproliferative and pro-apoptotic mechanisms. In preclinical models, it inhibited the angiogenesis and suppressed tumor growth of H22 hepatocellular carcinoma in murine models [70].

In human androgen-independent DU145 prostate cancer cells, PESV induces cell cycle arrest at the G1-phase by upregulating Kip1/p27 expression while downregulating cyclin E, thereby disrupting cyclin-dependent kinase (CDK) activity, a key driver of cancer progression, which is normally regulated by CDK inhibitors (CDKIs) [69,151]. This cell-cycle blockade is accompanied by apoptosis mediated through upregulation of the pro-apoptotic protein Bax and downregulation of the anti-apoptotic protein Bcl-2, highlighting PESV’s dual regulatory impact on proliferation and survival pathways. Importantly, PESV demonstrates preferential cytotoxicity toward prostate cancer cells over normal prostate epithelial cells, highlighting its potential as a targeted therapeutic agent [69].

Although these findings support PESV as a promising candidate, its precise molecular targets and the identity of the active peptide(s) within the extract remain unresolved. Further studies across different prostate cancer models are needed to validate its efficacy and clarify the signaling cascades involved [69].

### 5.4. Membrane Disruption and Tumor Microenvironment

#### 5.4.1. Hyaluronidase BmHYA1 from *Buthus martensii* Karsch

The hyaluronidase BmHYA1, isolated from the venom of the scorpion *Buthus martensiKarsch* (BmK), is a promising anticancer agent that modulates the tumor microenvironment with no observed toxic side effects [37,38,39,40,41,152]. This enzyme specifically targets hyaluronic acid (HA), a major glycosaminoglycan in the extracellular matrix (ECM). In many cancers, elevated HA levels promote tumor progression by facilitating cell migration invasion, and metastasis, primarily through interactions with its cell surface receptor, CD44 [37,38]. This interaction is also linked to malignant phenotypes and the expression of oncogenic variants, such as CD44v6 [153,154].

The antitumor mechanism of BmHYA1 is directly linked to its enzymatic activity. By breaking down HA, BmHYA1 disrupts the critical HA-CD44 signaling axis, which in turn inhibits oncogenic pathways. For instance, in triple-negative MDA-MB-231 breast cancer cells, BmKYA1 was shown to downregulate the expression of CD44v6 [38]. Furthermore, the degradation of the ECM reduces the physical or interstitial pressure within solid tumors, which enhances the penetration and efficacy of co-administered chemotherapeutic drugs [38,41]. This dual ability to act as a standalone antitumor agent and as a chemosensitizer highlights the therapeutic potential of BmHYA1, although further studies are needed to optimize its clinical application.

#### 5.4.2. RK1 from *Buthus occitanus tunetanus*

RK1, a 14-amino acid peptide isolated from the venom of the Tunisian scorpion *Buthus occitanus tunetanus*, represents a promising anticancer peptide with distinctive pharmacological properties. Biochemical and functional characterization has revealed that RK1 exhibits potent antitumor activity by simultaneously inhibiting cancer cell proliferation, migration, and angiogenesis without manifesting significant cytotoxicity toward normal cells, making it potentially effective against metastatic cancer. Moreover, RK1 demonstrated remarkable efficacy against glioblastoma (U87) and melanoma (IGR39) cell lines, with its mechanism of action involving the disruption of tumor cell adhesion and suppression of vascular growth, as evidenced by the chicken chorioallantoic membrane (CAM) assay [155].

Emerging evidence also indicates that RK1 may interfere with integrin-mediated pathways, which are critical for tumor cell adhesion, metastasis, and angiogenesis. This dual disintegrin-like activity, particularly on α1β1 and αvβ3 integrins, further expands its therapeutic potential by targeting the tumor microenvironment [155]. Given these multifaceted mechanisms, RK1 could serve as a foundational scaffold for developing novel anticancer therapeutics, either as a standalone agent or in combination with existing treatment modalities.

#### 5.4.3. Vmct1 from *Vaejovis mexicanus*

Vmct1 is a 13-residue non-disulfide-bridged cationic peptide (NDBP) originally identified from the venom gland transcriptome of the scorpion *Vaejovis mexicanus*. With a net charge of +2, ~69% hydrophobicity, and an amphipathic α-helical conformation, the native peptide exhibits antimicrobial activity but lacks significant anticancer effects. However, synthetic analogs of Vmct1 with lysine substitutions (Vmct1-K) were designed to enhance cationicity and bioactivity, resulting in potent cytotoxicity against melanoma (B16-F10), breast (MCF-7), and cervical (HeLa) cancer cell lines (IC_50_: 3.4–6.2 µM), with minimal toxicity to nontumoral VERO and red blood cells [28]. Mechanistic studies suggest a membrane-lytic mode of action. Scanning Electron Microscopy (SEM) images of Vmct1-K-treated cells revealed blebbing, wrinkling, and cell shrinkage, while increased propidium iodide uptake confirmed membrane disruption [28]. Together, these findings highlight Vmct1-K as a rationally optimized peptide with selective anticancer potential via direct membrane targeting.

#### 5.4.4. AcrAP1/AcrAP2 from *Androctonus crassicauda*

AcrAP1 and AcrAP2 are non-disulfide bridged peptides (NDBPs) isolated from the venom of the Arabian scorpion *Androctonus crassicauda*, which exhibit selective antimicrobial activity against Staphylococcus aureus and Candida albicans, but exhibit no cytotoxic or antiproliferative effects on a panel of different human cancer cell lines. However, cationicity-enhanced analogs AcrAP1a and AcrAP2a, engineered by substituting polar residues with lysine, displayed significantly improved biological activity. These analogs exhibited potent antiproliferative effects against multiple human cancer cell lines, including lung adenocarcinoma (NCI-H460), breast carcinoma (MCF-7 and MDA-MB-435s), and prostate carcinoma (PC-3), with IC_50_ values ranging from 2.1 to 3.6 µM. Interestingly enough, AcrAP1a also induced a paradoxical proliferative response in H460 and PC-3 cells at nanomolar concentrations, suggesting potential concentration-dependent dual effects that warrant further mechanistic investigation [27]. The enhanced anticancer activity of the analogs appears to correlate with their amphipathic α-helical structures and increased net positive charge, promoting interactions with negatively charged components of cancer cell membranes. While direct evidence of membrane lysis was not provided, the observed cytotoxic effects are consistent with mechanisms reported for other cationic antimicrobial peptides. The rational design of AcrAP analogs achieved by substituting neutral residues with lysine, similar to Vmct1-K demonstrates how structural optimization can amplify anticancer effects while minimizing off-target toxicity [28]. In summary, these findings highlight the potential of venom-derived peptide templates for rational design of anticancer agents. The structural optimization of AcrAP1/AcrAP2 into cationic analogs demonstrates a viable strategy to enhance bioactivity while maintaining selective toxicity, although concentration-dependent paradoxical effects underscore the need for thorough preclinical evaluation.

#### 5.4.5. Pantinins 1-3 from *Pandinus imperator*

Pantinins 1-3 are a family of short, cysteine-free, α-helical cationic peptides identified via cDNA cloning from the African scorpion *Pandinus imperator* venom gland transcripts. Their amphipathic helices underlie broad-spectrum antimicrobial activity, including potent effects against multidrug-resistant bacteria such as vancomycin-resistant *Enterococcus* strains (VRE) [67], which provides a rationale to explore their repurposing in oncology. Beyond their antibacterial profile, synthetic pantinin analogs have been shown to interact selectively with negatively charged membranes, producing membranolytic activity against human cancer cell lines while sparing erythrocytes, thus demonstrating low hemolytic potential [68].

This mode of action aligns with a growing class of anticancer antimicrobial peptides (AMPs) that exploit electrostatic differences in membrane lipid composition between malignant and healthy cells. While pantinins are α-helical, their mechanism is conceptually parallel to that of engineered β-hairpin peptides such as SVS-1, which undergo membrane-induced folding into a cytotoxic conformation upon contact with tumor membranes [156]. Such comparisons highlight a broader principle: secondary structure plasticity, driven by the tumor microenvironment, underpins the selective cytotoxicity of many AMPs.

Recent reviews have highlighted the growing interest in antimicrobial peptides (AMPs) as dual-purpose agents with antimicrobial and anticancer potential [157]. Within this broader context, pantinins exemplify scorpion-derived AMPs that combine small size, favorable safety profile, and well-defined helical structure, making them attractive scaffolds for further optimization. Strategies such as enhancing cationicity or conjugating to delivery systems may improve their tumor specificity and therapeutic index.

### 5.5. Multifunctional Peptides

#### 5.5.1. Chlorotoxin from *Leiurus hebraeus* and Derivatives

Chlorotoxin (CTX) is a 36-amino acid peptide originally isolated and purified from the venom of *Leiurus hebraeus* (formely *L. quinquestriatus hebraeus*) [158]. Structural analysis by NMR revealed a compact tertiary structure comprising an a-helix packed against three antiparallel b-strands stabilized by four disulfide bonds features that underlie its remarkable stability and high affinity for tumor cell membranes [159,160]. CTX selectively binds to gliomas and other neuroectodermal tumors, with negligible affinity for normal tissues [47,49,161,162], making it a valuable scaffold for tumor targeting.

Proteomics strategies have been critical for defining the molecular interactors of CTX. Affinity-column pull-downs combined with mass spectrometry further demonstrated that matrix metalloproteinase-2 (MMP-2) as a primary binding partner of CTX and revealed its association with a membrane complex containing MT1-MMP, TIMP-2, and avb3 integrin, implicating CTX in the modulation of pericellular proteolysis and adhesion [47,163]. These proteomic analyses reinforced earlier electrophysiology-based findings that CTX blocks glioma-enriched ClC-3 chlorine channels involved in cell migration and cytoskeletal remodeling [164,165]. More recently, protein microarray profiling coupled with mass spectrometry validation identified cortactin, an actin-binding and Src kinase substrate linked to invasive cancers, as a novel CTX interactor, suggesting an additional mechanism for its anti-migratory effects and making it an important biomarker for invasive cancers [166]. Such mass spectrometry-based proteomics-driven discoveries have expanded the CTX interactome beyond MMP-2, ClC-3, and annexin A2, highlighting neuropilin-1 (NRP1) as another critical binding partner for tumor selectivity [167,168].

Recent proteomic surveys of Buthidae venoms have further contextualized chlorotoxin within the venom proteome. Using LC-MS/MS, Mabunda et al. (2025) [169] identified chlorotoxin and CTX-like peptides as major cysteine-rich peptide families in *Leiurus hebraeus* venom. Importantly, integrative functional assays linked these CTX-containing fractions to cytotoxic and anti-migratory effects in glioma and melanoma cells, thereby bridging venom proteomics with cancer pharmacology. These findings reinforce the evolutionary conservation and therapeutic potential of CTX and highlight the value of advanced proteomics in connecting peptide abundance with functional anticancer activity [169].

Functional proteomic fragment mapping has shown that the C-terminal residues (29–36) of CTX retain partial activity, selectively inhibiting migration without fully blocking invasion [166]. This finding suggests that distinct CTX structural motifs may differentially regulate cytoskeletal versus proteolytic pathways. Internalization studies using live-cell imaging confirmed clathrin-mediated uptake of CTX derivatives [170], linking receptor binding with intracellular trafficking.

Building on these proteomics-driven insights, chlorotoxin has recently been incorporated into the design of chimeric antigen receptor (CAR) T cells. CLTX-CAR T cells employ the chlorotoxin peptide as a tumor recognition domain, redirecting T cell cytotoxicity against glioblastoma cells through MMP-2–associated binding. In a first-in-human Phase I clinical trial (NCT04214392), Barish et al. (2025) [171] demonstrated the feasibility and safety of intracavitary administration of CLTX-CAR T cells in patients with recurrent glioblastoma. No dose-limiting toxicities or immunogenicity were observed, and three of four patients achieved transient stable disease with evidence of CAR T cell persistence and local cytokine induction in the tumor cavity. Although long-term clinical benefit remains to be established, this study represents the first clinical application of a venom-derived peptide as the targeting domain of a cellular immunotherapy and highlights the translational potential of CTX as a proteomics-validated tumor ligand [171].

From a translational perspective, the development of synthetic CTX derivatives is best exemplified by TM-601, a radiolabeled version of the peptide with iodine-131 (^131^I-chlorotoxin) that demonstrated tumor-specific uptake and prolonged retention in Phase I/II clinical trials in malignant glioma [172,173,174]. Despite promising initial results, the clinical development of TM-601 was not pursued, and its Phase II trial was ultimately terminated in May 2009 for reasons that were not publicly disclosed (ClinicalTrials.gov Identifier: NCT00683761) [175]. Beyond therapy, proteomics-guided bioconjugates have supported CTX applications in diagnostics and imaging. BLZ-100 (Tumor Paint), a fluorescent CTX conjugate, enables intraoperative visualization of gliomas [176]. Nanotechnology-based delivery systems, such as CTX/mApoE-modified liposomes or MiniCTX shuttles, further enhance blood–brain barrier penetration and selective tumor targeting [177,178].

#### 5.5.2. Maurocalcine from *Scorpio maurus palmatus* and Related Peptides

Maurocalcine (MCa), a 33-amino acid peptide isolated from the venom of *Scorpio maurus palmatus*, exhibits dual functionality as a cell-penetrating peptide and a modulator of intracellular calcium signaling, with promising applications in oncology [62,179,180,181]. Structural analyses revealed three disulfide bonds and a cationic surface that facilitates membrane translocation, while its binding to ryanodine receptors (RyR1) triggers Ca^2+^ release from the endoplasmic reticulum [62,182]. Further studies have mapped its interaction with RyR1’s cytoplasmic domain, identifying key residues, such as Lys20, critical for both receptor activation and cell penetration [62,63,64,183]. Interestingly enough, a short 9-amino acid derivative from MCa known as MCaUF1-9 showed very favorable cell-penetrating efficacy and may be used to specifically target cancer cell in vivo, due to its acidic pH that matches tumor acidic environments [184]. Moreover, MCa has been shown to overcome doxorubicin resistance in MDA-MB231 breast cancer cell line [65]. Another MCa derivative, Pt-1-DMCa, a platinum–maurocalcin conjugate, has been shown to induces apoptosis in human glioblastoma U87 cells through ROS-dependent modulation of the PI3K/AKT/OfoxO3a signaling pathway [185]. Furthermore, different studies have revealed novel applications for MCa in delivering therapeutic cargo to cancer cells [62,179,180,181,186]. Overall, MCa and its peptide derivatives’ ability to efficiently cross cell membranes have made it an attractive candidate for drug delivery systems.

#### 5.5.3. AsTs-1 from *Androctonus australis*

The tetrapeptide AaTs-1, isolated from the venom of *Androctonus australis*, acts as a selective antagonist of the formyl-peptide receptor-like 1 (FPRL-1) in glioblastoma cells, leading to increased p53 expression while suppressing ERK/p38/JNK signaling [117]. AaTs-1 inhibits proliferation of U87 cells (IC_50_ = 0.56 mM), approximately twice as effective as temozolomide (TMZ), and potentiates its action. The peptide blocks store-operated calcium entry (SOCE), affecting SOC channels and endoplasmic reticulum calcium release, without inducing apoptosis or cell cycle arrest, indicating a non-cytotoxic antiproliferative mechanism. Structural modeling predicts stable binding of AaTs-1 to FPRL-1, similar to known antagonists, confirming its receptor-mediated mode of action. These findings suggest AaTs-1 as a promising lead for the development of low-cost, targeted glioblastoma therapeutics, alone or in combination with TMZ [117].

## 6. Immunomodulatory Effects

Scorpion venom contains a diverse array of bioactive peptides that can modulate both innate and adaptive immune responses. These peptides have been shown to influence immune cell function, including the repolarization of immune cells and the enhancement of antigen-specific responses.

### 6.1. Innate Immune Response Modulation

Proteomic analyses have revealed that scorpion venoms contain diverse proteins and peptides that modulate innate immune responses relevant to cancer. Several biochemical and mass spectrometry-based proteomic analyses have revealed that scorpion venom contains peptides and proteins with potential bioactive properties, including immunomodulatory effects, which can influence innate immune cells. Cota-Arce and colleagues have identified several proteins derived from the *Centruroides limpidus* venom, including neurotoxins, metalloproteases, phospholipases, hyaluronidases, and antimicrobial peptides. Among them, cancer inhibitory fractions CIF8 and CIF9 were able to induce the anti-inflammatory IL10 while suppressing the pro-inflammatory IFN-g in CD4+ T cells via Ca2+ channel modulation [187]. Moreover, they trigger innate immune crosstalk by activating macrophages and dendritic cells, evidenced by elevated IL-12 and TNF-α in co-cultures. IL-10 suppression could counteract tumor-associated immunosuppression, while Ca2+ channel-targeting toxins may disrupt cancer cell signaling [187]. Overall, this study demonstrated that these venom-derived proteins can shift the immune response toward Th1, Th2, or Th17 profiles, indicating their potential as modulators of T cell-mediated immunity that may be able to further explore their role in immune regulation relevant to cancer immunotherapy.

Another venom component, the peptide Css54 isolated from *Centruroides suffusus suffusus* has demonstrated significant immunomodulatory effects. Css54 enhances macrophage phagocytic activity while reshaping cytokine production. It suppresses IL-6, increases the anti-inflammatory IL-10, and modestly elevates IL-12p70 and TNF-α, with minimal impact on IFN-γ. This balanced immunomodulation suggests an ability to promote antimicrobial defense while dampening excessive inflammation [188]. These results position Css54 as a dual-function peptide with potential therapeutic applications in infection control and immune regulation, and they further raise the prospect of exploiting Css54 to modulate the tumor microenvironment by reinforcing innate immune mechanisms while preventing chronic inflammation

The venom’s ability to modulate toll-like receptor (TLR) signaling pathways is particularly noteworthy for its potential initial immune response against cancer cells. The peptide Ts1 isolated from *Tityus serrulatus* venom activates TLR2, TLR4, and CD14 on macrophages. This triggers MyD88-dependent NF-kB activation and TLR4-dependent, MyD88-independent c-Jun activation, as well as engagement of the ERK1/2 and p38 MAPK pathways, culminating in the release of TNF-α and IL-6 [189].

Venom-induced immune modulation has emerged as an important mechanism with potential application in cancer therapy. Several studies demonstrate that scorpion venoms can regulate macrophage polarization, shifting cells between pro-tumorigenic M2 and anti-tumorigenic M1 phenotypes [190]. In particular, venom from *Heteroctenus junceus* (previously known as *Rhopalurus junceus*) was shown to modulate pro-inflammatory cytokine production in F3II mouse mammary tumor cells, significantly reducing IL-6 and IL-1β while elevating TNF-α and IL-12, consistent with interference in the NF-κB pathway [191]. Oral administration of *H. junceus* venom in F3II tumor-bearing mice suppressed tumor growth and decreased serum TNF-α levels, indicating systemic immunomodulation [192]. These dual effects on cytokine regulation suggest that venom-derived molecules may repolarize tumor-associated macrophages from an M2- to an M1-like state, thereby reprogramming the tumor microenvironment toward an antitumor phenotype. Supporting this concept, studies with *Tityus serrulatus* venom show that macrophage responses are mediated by TLR2/TLR4/CD14 recognition, engaging NF-κB-linked pathways, which can be negatively modulated by peroxisome proliferator activated receptor-gamma (PPAR-γ) activity [189,193]. Collectively, these findings highlight scorpion venom as a source of immunomodulatory peptides with the potential to enhance antitumor immunity through macrophage reprogramming.

In another study, the toxic fraction FTox-G50 from *Androctonus australis* hector venom promoted M1 macrophage polarization in adipose tissue, increasing nitric oxide NO production and upregulating IL-12p40, IL-23, and NOS2 while suppressing M2 markers Arg1 and IL-10, in a TNF-α-dependent manner [194].

This modulation of innate immunity shows potential in cancer treatment, as it can help overcome the immunosuppressive tumor microenvironment and promote antitumor responses without causing systemic toxicity.

### 6.2. Adaptive Immunity Enhancement

The impact of scorpion venom on adaptive immunity has emerged as a crucial area of investigation in cancer immunotherapy. On one hand, scorpion venom can act as an adjuvant, promoting Th1-type adaptive immunity and increasing antibody avidity, which may improve the neutralizing capacity of antibodies against systemic threats. For example, the scorpion venom from *Hottentotta rugiscutis* exhibits adjuvant properties by enhancing adaptive immunity, specifically promoting hepatitis B surface antigen (HBsAg)-specific Th1 responses and increasing antibody avidity, which may improve neutralizing capacity against pathogens. This effect is mediated through neuroendocrine–immune interactions, including upregulation of nerve growth factor (NGF) and corticosterone (CORT), leading to activation of splenocytes and sustained IL-1β production [195].

Conversely, certain scorpion venom peptides selectively suppress pathogenic effector memory T lymphocytes by blocking Kv 1.3 potassium channels. For instance, HsTX1 from *Heterometrus spinnifer* is a potent Kv1.3 inhibitor (IC_50_ ≈ 12 pM) [55]. Likewise, Vm24 from *Vaejovis mexicanus smithi* exhibits extremely high specificity and affinity for Kv1.3 (Kd ≈ 2.9 pM, ~1500-fold selectivity), attenuating human CD4^+^ T_EM cell activation, cytokine production, and proliferation upon stimulation [80,196,197].

Recent studies identified Cm28, a peptide from *Centruroides margaritatus*, as a high-affinity inhibitor of Kv1.2 and Kv1.3 channels. In human CD4^+^ effector memory T cells, Cm28 suppressed activation markers such as IL-2R (CD25) and CD40L, underscoring its potential as an immunomodulatory agent [50].

This mechanism, classically explored in autoimmunity, may also be exploited in cancer, where Kv1.3 contributes to shaping the tumor–immune interface. In head and neck cancer, Kv1.3high CD8^+^ tumor-infiltrating lymphocytes have been identified as functionally competent effectors [198]. Supporting this rationale, margatoxin (MgTx), another Kv1.3 inhibitor from *C. margaritatus*, has demonstrated both in vitro and in vivo efficacy against A549 human lung adenocarcinoma, where it suppressed tumor cell proliferation, modulated cell-cycle regulators, and significantly reduced tumor growth in xenograft models [61]. These findings suggest that Kv1.3 inhibition may reprogram immune responses within the tumor microenvironment, potentially reducing Treg activity while preserving cytotoxic T cell function.

Beyond Kv1.3-targeting peptides, other venom-derived molecules contribute to adaptive immune modulation. For example, peptides AK and GK from *Buthus martensii* suppress TNF-α/EGFR/STAT3 signaling pathway, downregulating pro-inflammatory cytokines such as TNF-α and oncogenic drivers such as *c-Myc* while upregulating tumor suppressors (*p53*/*PTEN*). In gastric cancer models, these effects may help reverse immune evasion and restore antitumor immunity [32]. Collectively, these mechanisms underscore the potential of scorpion venom peptides to act either as immune adjuvants or as selective immunomodulators, thereby broadening their applicability in cancer immunotherapy.

### 6.3. Cytokine Profile Alterations

The manipulation of cytokine profiles by scorpion venom components represents a significant breakthrough in understanding their therapeutic potential. Different scorpion venoms are able to modulate cytokine networks through specific toxins, influencing both pro- and anti-inflammatory responses. FTox-G50 from *Androctonus australis hector* was shown to polarize adipose tissue macrophages (ATMs) toward an M1 phenotype, characterized by upregulated IL-12p40, IL-23, and NOS2 (iNOS) expression, while suppressing M2 markers such as Arginase-1 (Arg1) and IL-10. This shift is TNF-α-dependent, as demonstrated by the reversal of M1 gene expression upon etanercept (TNF-α antagonist) treatment [194]. Similarly, *Tityus serrulatus* venom (TsV) triggers systemic release of IL-1β, IL-6, TNF-α, and IFN-γ in severe envenomation cases [199]. In contrast, *Leiurus macroctenus* venom exhibits an atypical immunomodulatory profile, reducing pro-inflammatory cytokines (IL-6, IL-8, IL-1β) while elevating anti-inflammatory IL-4, IL-10, and IFN-γ in rat lungs, suggesting species-specific immune reprogramming [200]. The antimicrobial peptides BmKn1, BmKn2, and BmKn2-7 from *Mesobuthus martensii* further illustrate this duality, dampening TNF-α and IL-1β in *Litopenaeus vannamei* shrimp infected with *Vibrio parahaemolyticus*, while enhancing immune enzymes like phenoloxidase and complement component C3 [201]. For instance, proteomic analyses of *Hottentotta saulcyi* venom reveal a complex composition dominated by Na^+^- and K^+^-channel-targeting peptides, as well as a substantial lipid component (~1.2% dry weight), characterized via LC-MS/MS [52].

These findings underscore the potential of venom components to recalibrate immune responses, though their clinical translation requires further exploration of dose-dependent effects and signaling pathways.

### 6.4. Immune Cell Activation and Regulation

Scorpion venom components selectively target immune cells, influencing their activation and functional polarization. For example, the TzII and TzIII fractions of *Tityus zulianus* venom selectively activate human neutrophils, inducing PKC-dependent ROS production, an effect blocked by PKC inhibition [202]. Mass spectrometry has proven indispensable in characterizing such fractions: recent MALDI-TOF and LC-MS/MS analyses revealed that specific peptide components remain non-neutralized after antivenom interaction, as shown in *Odonthobuthus doriae* venom [203]. These findings underscore that non-neutralized low-molecular-weight peptides may persist and retain biological activity, providing a molecular basis for their selective engagement with neutrophil membranes. However, this effect is cell-type-specific, as eosinophils show negligible respiratory burst compared to neutrophils, reflecting distinct localization and regulatory mechanisms of the NADPH oxidase complex [126]. In macrophages, the β-toxin Ts1, also known as Tsγ, from *Tityus serrulatus* venom is recognized by TLR2, TLR4, and CD14, leading to MyD88-dependent NF-κB activation and triggering the release of IL-6, TNF-α, and nitric oxide (NO) [189,204]. Ts1 has been previously sequenced as a 61-amino-acid β-toxin, confirming its identity and structural class [205]. Conversely, Meuk7–3 from *Mesobuthus eupeus* suppresses effector memory T cells by blocking Kv1.3 channels, a strategy relevant for autoimmune diseases [206]. The Scorpine peptide from *Pandinus imperator* exemplifies dual functionality, activating phagocytes while exhibiting antimicrobial properties, though its precise immune targets remain under investigation [207]. Proteomic gaps persist, for several scorpion venoms, including *Androctonus crassicauda*, despite its medical importance and demonstrated biological activities. These studies highlight the potential of venom-derived peptides to modulate immune cell function, with implications for treating inflammatory disorders and cancer.

### 6.5. Potential for Immunotherapy Enhancement

Scorpion venom peptides offer promising avenues for cancer immunotherapy by targeting hallmarks of malignancy. Chlorotoxin (CTX) from *Leiurus hebraeus* binds glioma-specific chloride channels and MMP-2, inhibiting metastasis and enhancing blood–brain barrier penetration for drug delivery [47,146]. Its 4 kDa structure, confirmed by MALDI-TOF MS, underpins its clinical use in tumor imaging, such as CTX-Cy5.5 for fluorescence-guided surgery [43,47]. Similarly, BmK-AGAP from *Buthus martensii* Karsch exhibits analogous antitumor and immunomodulatory activities, though its characterization extends beyond reverse-phase high-performance liquid chromatography (RP-HPLC) to include genomic and functional analyses [208].

The Kv1.3 blockers, such as Meuk7–3 from *Mesobuthus eupeus*, demonstrate translational potential by suppressing autoreactive T cells in autoimmune diseases, with MALDI-TOF and structural modeling confirming their ion channel-targeting motifs [206].

*Androctonus crassicauda* venom and its peptides further expand this repertoire, inducing apoptosis and cell-cycle arrest in MCF-7 cells, with caspase-3 involvement; transcriptomic and proteomic studies of *A. crassicauda* venom glands detail the underlying peptide diversity [27,116,129,208,209,210,211].

Challenges include scalable production, addressed via heterologous expression in Escherichia coli, and precise immune modulation to increase bioactivity capabilities and avoid cytokine storms [208].

Together, these components highlight the dual utility of venom peptides: as direct cytotoxic agents and as immune modulators to enhance checkpoint inhibitor therapies.

## 7. Diagnostic Applications

Scorpion venom-derived peptides have emerged as innovative tools for tumor marking and imaging, owing to their high affinity for cancer-specific targets. Among them, chlorotoxin (CTX) has been extensively studied as a tumor-targeting agent due to its ability to selectively bind glioma and other tumor cells. Building on this specificity, synthetic derivatives such as BLZ-100 (tozuleristide, Tumor Paint) have been developed by conjugating CTX to fluorescent dyes, enabling real-time intraoperative visualization of tumor margins.

Preclinical studies demonstrated that CTX-based probes selectively label glioma, medulloblastoma, sarcoma, prostate, and colorectal tumors in mouse models, with near-infrared fluorophores offering optimal intraoperative imaging [212,213]. Comparative oncology trials in dogs confirmed safety and effective tumor visualization [214], highlighting the translational value of this approach [215]. Toxicology studies in multiple species, including rodents, dogs, and non-human primates, further established a favorable safety profile [216].

In humans, BLZ-100 has progressed to Phase 2/3 trials for pediatric central nervous system tumors, demonstrating safe administration and improved surgical precision through fluorescence-guided resection. Beyond gliomas, BLZ-100 has shown promise in identifying other lesions, such as cerebral vascular malformations [217], as well as non-melanoma skin cancers and melanoma, where it achieved accurate tumor imaging at clinically feasible doses [218]. These results underscore the potential of CTX-based probes to improve tumor visualization across diverse cancer types while maintaining low systemic toxicity.

### 7.1. Early Detection Methods

For early detection of *Hottentotta tamulus* (syn. *Mesobunthus tamulus*) envenomation, mass spectrometry-based proteomics was able to identify a rich repertoire of low-molecular-mass toxins, particularly Na+ and K+ ion-channel toxins that comprise a majority of the venom proteome [219]. To improve detection sensitivity in plasma, acetonitrile precipitation has been used to enrich these low-abundance peptides while simultaneously depleting high-molecular-weight plasma proteins, thereby facilitating their identification by mass spectrometry [220,221].

Although the direct application of gold nanoparticle (AuNP)–antibody conjugates and localized surface plasmon resonance (LSPR) biosensors to scorpion venom toxins has not yet been reported, AuNP-based platforms are well established in clinical diagnostics for providing rapid (often <10 min) colorimetric readouts and picomolar sensitivity for diverse biomolecules [222,223,224]. This suggests that similar systems could, in principle, be adapted for the ultrasensitive detection of scorpion venom peptides in envenomed patients. Such an approach would not only accelerate diagnosis in clinical settings but also establish a foundation for broader applications in oncology and precision medicine.

The specificity of venom peptide recognition, exemplified by Tamapin, a K^+^ channel toxin targeting SK2 [225], and α-neurotoxins acting on Na^+^ channels, underscores the potential of scorpion toxins as molecular probes. In oncology, chlorotoxin from *Leiurus hebraeus* provides a precedent: it binds selectively to tumor-associated receptors such as MMP-2 in gliomas and has been widely employed in tumor-targeting bioconjugates for both imaging and therapeutic applications [163].

Beyond diagnostics, this dual-use paradigm offers therapeutic opportunities. Kv10.1 (Eag1), a voltage-gated K^+^ channel aberrantly expressed in many cancers, represents a validated pharmacological target [226,227,228]. Conjugation of venom-derived peptides such as chlorotoxin or venom-derived cell-penetrating peptides (CPPs) to nanocarriers—including AuNPs, chitosan, polyethylene glycol (PEG), and polyethylenimine (PEI)—has been shown to improve tumor-selective delivery and antitumor efficacy while minimizing off-target effects [229,230,231]. By bridging diagnostics and therapy, this translational framework highlights the untapped potential of scorpion venom peptides in oncology and points toward their integration into next-generation precision medicine [220].

### 7.2. Biomarker Development

In biomarker discovery, untargeted ultra-high-performance liquid chromatography in tandem with quadrupole time-of-flight mass spectrometry (UHPLC-QTOF-MS) metabolomic profiling of hepatocellular carcinoma (HCC)-bearing mice treated with scorpion venom peptide extract (PESV) from *Buthus martensii* revealed 111 significantly altered serum metabolites (48 in negative ion mode, 63 in positive ion mode), highlighting major disruptions in pathways such as aminoacyl-tRNA biosynthesis, amino acid metabolism, glutathione metabolism, protein transport, protein digestion and absorption, and cAMP signaling [232]. These findings position PESV-induced metabolic signatures as potential diagnostic and therapeutic biomarkers in HCC. The observed pathway-level perturbations suggest that scorpion venom peptides could modulate key metabolic networks relevant to tumor progression and therapy. Future studies should clarify whether these shifts intersect with known cancer-related mechanisms, including immune evasion, nitrogen metabolism, or drug resistance. Bridging these metabolomic insights with venom peptide pharmacology may open avenues for developing venom-based diagnostics and therapeutic strategies in precision oncology.

### 7.3. Therapeutic Monitoring

The neurotoxic properties of *Odontobuthus doriae* venom, which disrupts ion channels and neurotransmitter activity, also make it a promising candidate for diagnostic applications. Recent advances in plasmonic biosensing technology have enabled the detection of venom-induced neurotoxicity in human serum through circular dichroism (CD) measurements. By employing an achiral gold-coated plasmonic nanostructure, researchers demonstrated that minute changes in venom concentration alter the Stern layer’s action potential, shifting the sensor’s refractive index and producing measurable CD responses. This method achieved high sensitivity (27.4 at 520 nm) and rapid detection, offering a potential tool for diagnosing envenomation or monitoring venom-derived therapeutics. Such biosensors could be adapted to detect cancer-specific biomarkers, leveraging venom components’ selective binding properties to improve early diagnosis and personalized treatment strategies [233].

### 7.4. Integration with Current Diagnostic Tools

Integration with current diagnostic tools further expands the clinical potential of venom-based agents. For instance, chlorotoxin (CTX) has been combined with magnetic resonance imaging (MRI) and near-infrared fluorescence in preclinical studies for glioblastoma detection [47,234,235]. In addition, CTX–drug conjugates such as CTX–onconase have shown increased antitumor efficacy in glioma models compared to unconjugated mixtures [236], while CTX-decorated liposomal nanoparticles have effectively delivered siRNA or antisense oligonucleotides to glioblastoma cells in vitro and in vivo [237]. Despite these promising results, challenges remain in translating these approaches into routine clinical use, particularly regarding standardization, scalability, and regulatory approval. Nonetheless, the high specificity of CTX for tumor cells, its modular platform design, and its blood–brain-barrier permeability position it as a valuable adjunct to conventional diagnostic and therapeutic strategies for glioblastoma and related malignancies.

## 8. Drug Development and Delivery Systems

### 8.1. Peptide Modification Strategies

The development of peptide modification strategies for scorpion venom components represents a critical advancement in cancer drug development. Mass spectrometry analysis has identified key modification sites, such as disulfide bridges in chlorotoxin (CTX), which enhance structural stability and target specificity for glioma cells [238,239]. The polyethylene glycol conjugation (PEGylation) of CTX demonstrated that PEG-CTX conjugates significantly improved tumor-targeting efficiency, as evidenced by an 8-fold increase in glioma accumulation compared to unmodified CTX when delivered via polyamidoamine (PAMAM) dendrimer nanoparticles [240]. This modification not only enhanced systemic circulation but also preserved CTX’s binding affinity to matrix metalloproteinase-2 (MMP-2), a receptor overexpressed in gliomas, enabling precise intracellular delivery of therapeutic genes such as TRAIL (tumor necrosis factor-related apoptosis-inducing ligand) [240].

Moreover, recent studies with *Tityus stigmurus* venom-derived peptides demonstrate that PEGylation strategies can be extended to other venom components for cancer applications, as evidenced by polylactic acid-polyethylene glycol copolymer (PLA-PEG) encapsulation of Stigmurin analogs (S1 and S2), which reduced hemolysis by 20% while maintaining antiproliferative activity against macrophage-like RAW264.7 cancer cells, highlighting the dual benefit of improved safety and retained bioactivity in venom-based nanotherapeutics [241].

Cyclization of BmK peptides from *Buthus martensii* has been shown to enhance their therapeutic potential. In the case of Buthicyclin, a cyclic peptide derived from Defensin 4 (BmKDfsin4), the introduction of a disulfide bond between terminal cysteine residues improved structural stability and prolonged serum half-life [242]. The peptide’s high binding affinity to opioid receptors (Mu/Kappa/Delta types) and low toxicity profile (LD50 > 20 mg/kg, <5% hemolysis at 4 mg/mL) suggests its scaffold could be repurposed for targeting cancer-associated receptors, such as opioid growth factor receptor (OGFR), which is implicated in tumor proliferation and metastasis [242].

### 8.2. Nanoparticle-Based Delivery

Nanoparticle delivery systems have revolutionized venom peptide therapeutics by addressing challenges such as rapid degradation and poor tissue penetration while improving controlled release, stability, and cellular uptake, all with reduced toxicity [243]. Gold nanoparticles, for instance, have shown promise in cancer treatment due to their ability to be co-functionalized with targeting ligands like peptides and stabilizing agents such as polyethylene glycol (PEG), enhancing cellular uptake while maintaining stability under physiological conditions [229,230,231]. This approach reduces non-specific protein adsorption and immune clearance, positioning gold nanoparticles as promising candidates for targeted drug delivery and imaging in oncology [244].

The anticancer potential of scorpion venom has been further demonstrated through liposomal encapsulation of venoms from *Androctonus bicolor*, *Androctonus crassicauda*, and *Leiurus quinquestriatus*, which significantly enhances their anticancer efficacy against colorectal cancer cells (HCT-8). Venom-loaded liposomes exhibited superior stability, controlled release, and increased cytotoxicity compared to free venoms, as evidenced by reduced cell viability, elevated reactive oxygen species (ROS) generation, and induction of apoptosis and cell cycle arrest at the G0/G1 phase. This nano-delivery system not only improved the therapeutic index of the venoms but also minimized non-specific toxicity, underscoring its potential as a targeted and efficient strategy for cancer therapy [22].

Chitosan nanoparticles (CN) have emerged as another promising platform for venom-based cancer therapies. For example, CN enhanced antimicrobial activity of *Tityus stigmurus* venom (Tsv) through high encapsulation efficiency (>78%), stability, and controlled release, with a small particle size (<180 nm) and positive charge (+23 to +28 mV) that improves cellular interactions [245]. Similarly, Rebbouh et al. (2020) [246] demonstrated the efficacy of CN in detoxifying and delivering the neurotoxin Aah II from *Androctonus australis hector* venom. The nanoparticles achieved 96.66% encapsulation efficiency and a biphasic release profile (55% within 8 h and 80% over 5 days), significantly reducing toxicity while enhancing immunogenicity. Immunized mice survived lethal toxin doses (up to 8 LD50), highlighting the potential of CN for venom-based therapies [246]. 

These properties are particularly advantageous for antitumor applications, as chitosan’s biocompatibility and mucoadhesive nature can enhance tumor targeting and retention. By encapsulating venom peptides with known anticancer properties, CN could enhance therapeutic efficacy while minimizing systemic toxicity. Such an approach leverages the dual benefits of detoxification and controlled delivery, offering a promising strategy for developing targeted cancer therapies.

### 8.3. Targeted Delivery Systems

Targeted delivery systems have been refined through different experimental insights. For example, chlorotoxin CTX-conjugated iron oxide nanoparticles displayed significantly enhanced internalization in glioma cells, with approximately 50% higher uptake in C6 glioma cells compared to untargeted nanoparticles, supporting their role in improving tumor-specific delivery [247]. Beyond demonstrating selective binding and uptake, such systems also open opportunities for integration with proteomic strategies to profile nanoparticle-cell surface protein interactions, enabling a deeper understanding of receptor landscapes that mediate efficiency and refining future venom-based nanocarriers [248].

A particularly relevant innovation has been the development of dual-targeting systems designed to overcome both the blood–brain barrier (BBB) and the tumor microenvironment. In this context, Yue and colleagues engineered PEGylated liposomes conjugated with OX26, an antibody against the transferrin receptor, together with CTX, thereby coupling BBB transcytosis with glioma-specific targeting [249]. The dual-targeting lipoplexes successfully transported therapeutic plasmid DNA across the BBB in vitro and in vivo, decreased C6 glioma cell viability to nearly 46% in a co-culture BBB model and achieved marked tumor reduction and prolonged median survival (46 days versus 13 days in controls) in rat glioma models. Histopathological and immunohistochemical analyses confirmed that the therapeutic effect was mediated by targeted *hTERTC27* gene expression within the tumor site, underscoring the complementary roles of OX26 in mediating BBB transport and CTX in conferring tumor selectivity [249].

Collectively, these findings illustrate how venom-derived ligands such as CTX can be integrated into multifunctional delivery platforms to improve targeting precision and therapeutic efficacy. The combination of proteomic insights into receptor expression with rational design of dual-targeted nanocarriers may pave the way for next-generation scorpion venom–based therapeutics capable of addressing both systemic delivery challenges and the heterogeneity of tumor microenvironments.

## 9. Clinical Studies and Trials

### 9.1. Preclinical Studies

Extensive preclinical research has laid a robust foundation for the clinical translation of scorpion-venom–derived peptides. Venomics analysis combining transcriptomics and proteomics, has cataloged venom constituents with high resolution, allowing the identification of chlorotoxin (CTX)-like peptides with anticancer potential. Early venomic efforts provided the essential sequence-level starting points for functional studies [161,250]. Biochemical and structural analyses subsequently mapped CTX mechanisms of action, revealing direct interactions with tumor-associated membrane proteins, such as MMP-2 and NRP1. These interactions clarified the mechanism of cellular uptake and CTX retention within glioma models [47,168].

In vivo studies in rodents and other models reinforced these findings by demonstrating selective tumor binding, effective intratumoral localization, and pronounced antitumor activity of CTX-based agents [162,168]. Imaging-enabled preclinical studies with radiolabeled or fluorescent CTX conjugates tracked biodistribution and tumor localization in real time, thereby visually substantiating CTX’s tumor-targeting capabilities [49,251]. These cumulative preclinical data collectively supported first-in-human testing of CTX-derived agents [52,168], providing the rationale for translation into early-phase clinical trials.

Another significant preclinical development in this domain is the discovery and characterization of the peptide Gonearrestide, an anticancer peptide derived from the Moroccan fat-tail *Androctonus mauretanicus* scorpion venom. This peptide was identified using an integrated high-throughput platform combining next-generation transcriptome sequencing (NGS) and MS/MS proteomic analysis, which enabled the screening of over 200 novel peptides and rational functional selection [53]. Gonearrestide showed potent in vitro activity, inducing G1 cell-cycle arrest and inhibiting proliferation in human colorectal cancer cells. Mechanistically, it inhibited cyclin-dependent kinase 4 (CDK4) while upregulating key cell-cycle regulators p21, p27, and cyclin D3. Its antitumor activity extended to in vivo xenograft mouse models, confirming potent reduction in tumor growth, with minimal toxicity toward noncancerous epithelial cells and erythrocytes [53]. Systematic reviews further acknowledge Gonearrestide as a leading example of venom-derived peptides with translational potential [20].

Collectively, these investigations highlight how the synergy of molecular venomics, proteomic mapping, and imaging technologies accelerates bench-to-bedside development of scorpion venom-derived peptides. This multidisciplinary approach has not only identified promising excipient peptides such as CTX and Gonearrestide but also elucidated their mechanisms of action, helping bridge molecular discovery with early-phase clinical exploration.

### 9.2. Clinical Trials

Phase I clinical evaluation of scorpion venom-derived compounds has marked an important step toward their translation into oncology. Among these, chlorotoxin (CTX)–based constructs are the most extensively studied, with two agents: TM-601 and BLZ-100 (Tozuleristide), advancing into first-in-human clinical trials. These compounds were initially designed to assess safety, pharmacokinetics, and tumor-targeting or intraoperative imaging capabilities, and they represent distinct but complementary applications of venom peptides in oncology (Table 2).

#### 9.2.1. TM-601 (^131^I-Chlorotoxin Conjugate) 

TM-601 is a synthetic chlorotoxin analog radiolabeled with iodine-131 (^131^I), for targeted radionuclide delivery to gliomas. In a Phase I single-dose, open-label trial, 18 adult patients with recurrent high-grade gliomas received intracavitary administration of ^131^I-TM-601 following tumor resection [173]. The compound showed selective tumor uptake and prolonged intracavitary retention for up to 14 days, with minimal systemic distribution and rapid clearance of non-bound peptide within 24 to 48 h. Importantly, the therapy was well tolerated, and no dose-limiting toxicities were reported. Clinical outcomes were encouraging at 180 days post-treatment, several patients exhibited stable disease or partial radiographic responses, and two remained without evidence of disease for more than 30 months [173]. These findings confirmed the safety and tumor-targeting specificity of ^131^I-TM-601, providing the basis for subsequent multicenter Phase II studies that further validated its tolerability and suggested signals of tumor control in subsets of patients [167].

#### 9.2.2. BLZ-100 (Tozuleristide, “Tumor Paint”)

BLZ-100, also known as tozuleristide or “Tumor Paint,” represents a diagnostic and surgical innovation derived from chlorotoxin. It consists of chlorotoxin conjugated to a near-infrared fluorophore, enabling real-time intraoperative visualization of tumors [213]. In a Phase I dose-escalation trial in adults with newly diagnosed or recurrent gliomas, BLZ-100 demonstrated a favorable pharmacokinetic profile, absence of dose-limiting toxicities, and robust tumor-specific fluorescence that facilitated surgical delineation of tumor margins [176]. Beyond gliomas, early-phase evaluations have extended its application to breast carcinoma, neck squamous cell carcinoma, cutaneous malignancies, and cerebral vascular malformations where it has similarly provided intraoperative contrast and improved visualization of neoplastic and malformed tissue [217,218,252,253].

Taken together, these first-in-human studies illustrate the translational trajectory of scorpion venom–derived peptides from preclinical discovery to clinical application. TM-601 highlights the potential of chlorotoxin analogs for selective delivery of therapeutic radionuclides to malignant gliomas, while BLZ-100 demonstrates their utility as tumor-targeted imaging agents capable of enhancing surgical precision. Both exemplify how venomic and proteomic research can yield clinically valuable innovations at the interface of targeted therapy and image-guided oncology.

## 10. Current Challenges and Future Directions

### 10.1. Production and Scale-Up Issues

The production and scale-up of scorpion venom-derived therapeutics remain major challenges in their development as anticancer agents. The structural complexity of venom peptides, particularly their disulfide-rich scaffolds, makes large-scale synthesis and recombinant expression technically demanding. Recombinant systems, including bacterial, yeast, and insect cell hosts, have been explored to produce bioactive peptides, although maintaining proper folding and disulfide connectivity is a persistent obstacle [21,136]. Mass spectrometry and proteomic analyses have been essential for characterizing venom components and verifying recombinant products, confirming sequence integrity, disulfide patterns, and post-translational modifications. These analytical approaches support quality control and help ensure batch-to-batch consistency. However, true large-scale manufacturing remains difficult, and further advances in expression technologies, folding strategies, and purification methods are required before venom-derived peptides can be produced reliably for widespread clinical use.

### 10.2. Regulatory Considerations

The regulatory development of scorpion venom-derived therapeutics presents unique challenges, particularly related to product characterization, manufacturing consistency, and safety evaluation. The structural complexity of venom peptides, including their disulfide-rich scaffolds and post-translational modifications, necessitates sophisticated analytical methods to meet regulatory standards. Mass spectrometry and proteomic approaches have been valuable in supporting early-stage product characterization by providing detailed molecular fingerprints, verifying disulfide connectivity, and assessing purity. These analyses contribute to quality control and the establishment of critical quality attributes, which are central to regulatory submissions [21,136]. However, the integration of proteomic tools into regulatory processes for venom-derived therapeutics remains largely preclinical and supportive, rather than being mandated for safety monitoring or biomarker validation. Future regulatory pathways may benefit from standardized and validated proteomic methodologies; however, conventional analytical chemistry and pharmacology approaches currently remain the backbone of regulatory compliance.

### 10.3. Cost and Accessibility

The production and scale-up of scorpion venom-derived therapeutics face substantial hurdles that stem from the structural complexity and heterogeneity of venom peptides, many of which are disulfide-rich and require precise folding and post-translational features to retain activity. As a result, development programs increasingly favor synthetic or recombinant routes over direct venom extraction, but these approaches introduce manufacturing challenges in expression, folding, and downstream purification. Proteomics-driven venomics centered on high-resolution LC-MS/MS has become indispensable for defining critical quality attributes (identity, purity, isoform composition, and stability) and for establishing batch comparability, thereby informing process optimization and release testing. These analytical requirements, together with the need for controlled production of correctly folded peptides, contribute to the overall complexity and cost of goods, and they help explain why only a few scorpion-toxin candidates (e.g., chlorotoxin derivatives) have advanced clinically while no scorpion-venom drug has yet reached the market. Current literature highlights these scientific and technological bottlenecks and points to ongoing advances in recombinant expression, folding strategies, and high-throughput venomics as levers to improve manufacturability and scalability, though further optimization and standardization will be needed before routine, cost-efficient production is achieved [20,254,255,256].

### 10.4. Research Gaps

Comprehensive venomics has clarified that major knowledge gaps still limit the development of scorpion venom–derived therapeutics for cancer. Large portions of the venom peptidome remain insufficiently characterized across species, with unresolved questions about isoform diversity, post-translational modifications, and their impact on stability and bioactivity [81,87,169,203,257,258,259,260,261,262,263]. Proteomics and transcriptomics have mapped candidate toxins and hinted at complex mechanisms, yet for lead bioactive peptides such as chlorotoxin, target engagement and downstream pathways remain debated, underscoring the need for rigorous mechanism-of-action studies in relevant cancer models and microenvironments [47,163,264,265,266]. Furthermore, translational gaps persist: there are no validated proteomic biomarkers for patient selection, pharmacodynamic readouts, or resistance monitoring in clinical settings, reflecting the absence of efficacy-focused trials to generate such data. While [^131^I]-TM-601 has undergone Phase I-II clinical trials for recurrent glioma, these studies were primarily safety-oriented and did not include broad biomarker development [171,173]. Addressing these gaps will require standardized, cross-lab venomics workflows, including top-down and de novo analysis, to define critical molecular features. Systematic studies of PTMs and structure–function relationships, and integrated preclinical pipelines that couple proteomics with functional assays to prioritize truly druggable venom leads [43,81,168,262,267,268].

### 10.5. Future Research Directions

The future development of scorpion venom-derived cancer therapeutics holds significant promise, with proteomic and transcriptomic analysis continuing to guide discovery and preclinical evaluation. Venomics approaches integrating mass spectrometry-based proteomics and sequencing have expanded the catalog of bioactive peptides, enabling the identification of novel components with potential therapeutic applications [81]. Efforts to improve specificity and efficacy increasingly rely on structural and proteomic characterization of toxin–target interactions, exemplified by ongoing studies on chlorotoxin derivatives [169]. Advances in delivery systems, including nanoparticle formulations and peptide conjugates, are being investigated to improve stability and tumor selectivity. Although not yet widely applied to scorpion toxins, nanoparticle delivery of venom-derived peptides in broader venom research demonstrates proof-of-concept potential [269].

Looking ahead, the integration of artificial intelligence with proteomic datasets offers a path toward accelerating venom peptide discovery and prioritizing candidates with favorable properties [270,271]. Combination therapies also represent a promising avenue, as proteomic profiling can reveal synergistic effects between venom peptides and established anticancer agents [43]. Finally, the application of proteomic biomarkers to stratify patients and guide treatment design remains largely speculative, but it represents an important goal for developing personalized venom-derived therapeutics. Collectively, these strategies are expected to produce more effective and targeted interventions while addressing current translational limitations.

## 11. Conclusions and Future Perspectives

Scorpion venom represents a highly diverse natural library of peptides and proteins with unique structural and functional characteristics, making them valuable candidates for anticancer drug development. Proteomic advances have been pivotal in uncovering this molecular diversity, enabling the identification of novel bioactive compounds and mapping their mechanisms of action at unprecedented resolution. These peptides have demonstrated the capacity to selectively target ion channels, disrupt tumor-supportive signaling pathways, induce apoptosis, and remodel the tumor microenvironment while often sparing healthy cells. In addition, the capacity of some venom-derived peptides, such as chlorotoxin, to cross the blood–brain barrier highlights their translational potential in targeting aggressive and otherwise inaccessible tumors.

Despite these advances, important challenges remain. The clinical development of venom-derived therapeutics is hindered by obstacles such as large-scale peptide synthesis, in vivo stability, pharmacokinetics limitations, delivery strategies, and the risk of off-target effects. Addressing these barriers will be essential to enable the translation from preclinical to clinical application. In this regard, the future of scorpion venom-based drug discovery is closely linked to the evolution of proteomic technologies. Single-cell venom gland proteomics promises to delineate precise cellular origins of toxins, uncovering rare peptides and clarifying the understanding of the regulation of venom biosynthesis. Top-down proteomics will allow comprehensive characterization of intact venom proteins and their native post-translational modifications, providing insights into proteoforms that directly correlate with biological activity. Furthermore, interactomics will be key to systematically mapping the binding partners of venom peptides within cancer cells, validating molecular mechanisms, and identifying opportunities for rational combination therapies.

Beyond discovery, proteomics will play a pivotal role in addressing translational challenges. Advanced mass spectrometry will ensure batch-to-batch consistency during manufacturing by confirming sequence integrity, disulfide connectivity, and overall purity. Furthermore, proteomic data will guide the rational design of optimized analogs with improved stability. reduced immunogenicity, and improved pharmacological profiles through strategies such as peptide cyclization and PEGylation. The integration of proteomic biomarkers into preclinical and clinical settings could also help stratify patients, monitor therapeutic efficacy, and quickly identify mechanisms of resistance.

Looking ahead, the integration of proteomics with transcriptomics, structural biology, computational modeling, and synthetic biology will be crucial to optimizing venom-derived therapeutics. Strategies such as nanoparticle conjugation or imaging probes, and targeted delivery strategies may enhance efficacy and minimize off-target effects. Moreover, the immunomodulatory and tumor-microenvironment–modulating properties of certain venom peptides highlight promising synergies with immunotherapies and standard treatments.

In summary, scorpion venom remains an underexploited source of bioactive molecules with significant potential in oncology. By combining high-throughput omics technologies, nanotechnology, medicinal chemistry, artificial intelligence, and translational research may transform these natural peptides into clinically relevant anticancer agents. Realizing this promise will require sustained interdisciplinary collaboration to overcome current barriers and accelerate innovation. While significant progress has been made, key proteomic gaps remain, particularly in the detection of low-abundance peptides, characterization of post-translational modifications, and integrating with other omics layers such as transcriptomics and structural biology. Future advances in multi-omics and computational approaches will be critical to translating scorpion venom peptides into effective therapeutic tools against cancers resistant to conventional treatment.

## Figures and Tables

**Figure 1 ijms-26-09907-f001:**
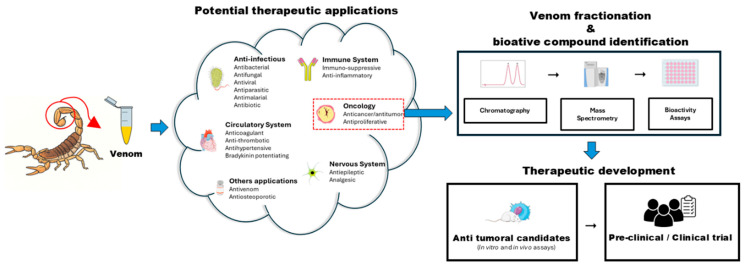
A schematic overview of the proteomics-driven workflow for the discovery of bioactive compounds from scorpion venom. The process begins with the extraction of crude scorpion venom, which is then fractionated using HPLC or SEC to reduce its complexity. Each fraction is analyzed by liquid chromatography–tandem mass spectrometry (LC-MS/MS) for peptide sequencing. The resulting mass spectrometry data are queried mainly against UniProt and VenomZone databases and processed using bioinformatics tools to identify the venom’s constituent proteins and peptides. This systematic approach is the crucial in identifying novel bioactive peptides that can serve as leads for the development of new anticancer agents. Figure generated using Microsoft PowerPoint.

**Figure 2 ijms-26-09907-f002:**
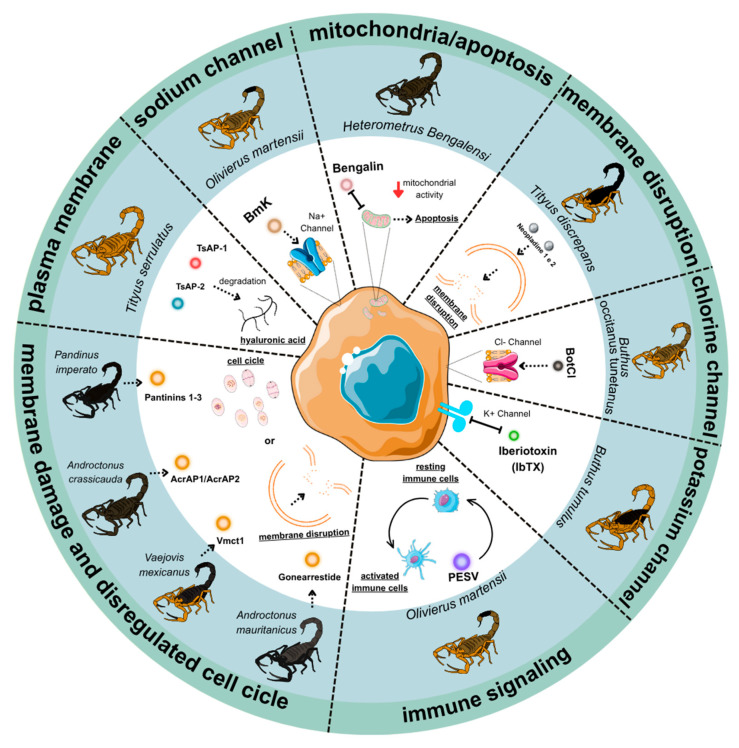
Mechanisms of Anticancer Action of Selected Scorpion Venom Peptides. Bioactive peptides from different scorpion species target key cellular processes, including membrane disruption, ion channel modulation, apoptosis induction, immune signaling, and cell cycle regulation. These mechanisms contribute to their antitumor effects and support their potential as therapeutic agents. Figure generated using Microsoft PowerPoint.

**Table 1 ijms-26-09907-t001:** Scorpion venom-derived peptides and proteins with demonstrated anticancer activity.

Peptide/Protein	Scorpion Source	Molecular Mass/kDa	Molecular Target	Endpoint (Mechanism, IC50)	Primary Cancer Model(s)	Proteomic Validation	Key References
AcrAP1	*Androctonus crassicauda*	8.7	Cell membranes	Blocks recognition and binding of target DNA by the Cascade complex.	Prostate, Lung	NR	[22,26,27,28]
AcrAP2		8.7					
AGAP-SYPU2	*Buthus martensii* Karsch	7.3	Voltage-gated sodium channels (Nav1.4, Nav1.5, Nav1.7)	Induces apoptosis and inhibits proliferation and migration.	Fibrosarcoma, Colon	Yes (MALDI-TOF-MS)	[29,30,31]
AK and GK	*Buthus martensii* Karsch	-	TNF-α/EGFR/STAT3 signaling pathway	Downregulates pro-inflammatory cytokines (TNF-α) and oncogenes (c-Myc); upregulates tumor suppressors (p53/PTEN)	Gastric cancer	NR	[32]
Bengalin	*Heterometrus bengalensis*	2.2	Mitochondria (Bax/Bcl-2)	Induces intrinsic apoptosis via mitochondrial depolarization and cytochrome c release.	Leukemia	NR	[33,34,35,36,37]
BmHYA1	*Buthus martensii* Karsch	-	Hyaluronic Acid (HA) in the extracellular matrix	Degrades HA, disrupting the HA-CD44 signaling axis; reduces tumor interstitial pressure, enhancing drug penetration	Breast	NR	[38,39,40,41]
BmK crude venom	*Buthus martensii* Karsch	Mixture	PI3K/Akt signaling, PTEN/p27	Inhibits oncogenic signaling and induces apoptosis.	Glioma, Lymphoma, Breast, Hepatoma	NR	[42,43]
BmKn2	*Mesobuthus martensii* Karsch	8.0	Mitochondria (Bax/Bcl-2)	Induces apoptosis, ROS generation, and mitochondrial dysfunction.	Oral Squamous Carcinoma	NR	[44,45]
BotCl	*Buthus occitanus tunetanus*	-	ClC-3, MMP-2	A chlorotoxin-like peptide that inhibits migration and invasion.	Glioblastoma, Breast	NR	[46,47,48]
Chlorotoxin (CTX)	*Leiurus hebraeus*/*quinquestriatus*	4.0	ClC-3 chloride channels, MMP-2	Inhibits tumor cell invasion and migration.	Malignant Glioma	NR	[47,49]
Cm28	*Centruroides margaritatus*	2.8	Potassium channel Kv1.3	Suppresses T cell activation markers (IL-2R and CD40L).	Not tested	NR	[50]
Crude venom	*Hottentotta saulcyi*	-	Na+ and K+ channels	Induces apoptosis in MCF-7 cells via TNF-α and caspase-3 upregulation.	Breast, Colon Cancer	Yes (LC-MS/MS)	[51,52]
Gonearestide	*Androctonus mauritanicus*	-	Cell membranes, CDK regulators	Induces G1 cell cycle arrest by downregulating CDK4 and upregulating p21/p27.	Prostate, Lung, Colorectal	Yes (LC-MS/MS, MALDI-TOF)	[53]
Hemiscorpius lepturus crude venom	*Hemiscorpius lepturus*	Mixture	Bax, p53, caspase-3 activation, Bcl-2 suppression	Induces apoptosis with low toxicity to normal cells.	Colorectal Cancer	NR	[54]
HsTX1	*Heterometrus spinnifer*	3.8	Potassium channel Kv1.3	Suppresses pathogenic effector memory T-lymphocytes.	Not tested	Yes (MALDI-TOF)	[55]
Iberiotoxin (IbTX)	*Hottentotta tamulus* (*syn. Buthus*/*Mesobuthus tamulus*)	4.3	Potassium channels Kv1.1, Kv1.3	Blocks K^+^ currents, leading to Ca^2+^ dysregulation and apoptosis.	Breast Cancer	NR	[56,57,58,59,60].
Margatoxin (MgTX)	*Centruroides margaritatus*	4.2	Potassium channel Kv1.3	Suppresses proliferation and inhibits tumor growth.	Lung Adenocarcinoma	NR	[61,62,63,64]
Maurocalcine (MCa)	*Scorpion maurus palmatus*	3.9	Ryanodine receptor (RyR1)	Induces Ca^2+^ release; acts as a cell-penetrating peptide for drug delivery.	Breast Cancer	NR	[62,63,64,65]
Neopladine 1 and 2	*Tityus discrepans*	-	Fas ligand → caspase-8 → Bid/tBid	Induces extrinsic apoptosis via the death receptor pathway.	Breast Cancer	Yes (MALDI-TOF)	[66]
Pantinins 1	*Pandinus imperator*	7.9	Cell membranes	Destabilizes and disrupts the membrane of target cells.	Breast, Prostate Adenocarcinoma	NR	[67,68]
Pantinins 2		7.7					
Pantinins 3		7.7					
PESV	*Buthus martensii* Karsch	Mixture	Cyclin E, Kip1/p27, Bax/Bcl-2	Induces G1-phase cell cycle arrest and apoptosis	Hepatocellular carcinoma, Prostate	NR	[69,70]
Smp43	*Scorpio maurus palmatus*	9.1	Mitochondria	Induces apoptosis by activating caspase-1, triggering pyroptosis.	Lung Cancer	NR	[71,72,73,74]
Smp24	*Scorpio maurus palmatus*	8.9	Mitochondria	Induces apoptosis via mitochondrial membrane potential reduction and ROS production.	Lung Cancer	NR	[75,76]
TsAP-1	*Tityus serrulatus*	8.4	Cell membranes	TsAP-2 IC_50_ = 4 μM (SKBR3 cells); Induces apoptosis through membrane disruption	Lung, Prostate, Squamous Carcinoma	Yes (MALDI-TOF/TOF)	[77,78,79]
TsAP-2		8.4					
TsAP-S2	*Tityus serrulatus*	1.7	Mitochondria	TsAP-S2 IC_50_ = 0.83 μM. Induces apoptosis through mitochondrial disruption.	Not tested	Yes (ESI-MS)	[78];
Vm24	*Vaejovis mexicanus smithi*	3.9	Potassium channel Kv1.3	Suppresses pathogenic effector memory T-lymphocytes.	Breast Cancer	Yes (LC-MS/MS, MALDI-TOF)	[80]

NR, not reported in the original source; -, data not available. MALDI-TOF, Matrix Assisted Laser Desorption and Ionization–Time of Flight

**Table 2 ijms-26-09907-t002:** Clinical trials of scorpion venom-derived compounds.

Compound Name	Trial Phase	Condition	Key Findings	Status
Chlorotoxin (CTX)/TM-601	Phase I/II	Recurrent high-grade glioma (glioblastoma, anaplastic astrocytoma)	Selective tumor binding; safe intratumoral use; prolonged tumor retention; limited systemic toxicity	Completed Phase I/II; discontinued for glioma therapy, redirected to imaging and drug-delivery applications
BLZ-100 (Tumor Paint, Tozuleristide, Chlorotoxin–ICG conjugate)	Phase I/II	Pediatric and adult brain tumors, breast cancer, sarcoma	Enabled real-time intraoperative tumor visualization; improved surgical margin delineation; well tolerated	Ongoing Phase II (pediatric brain tumors, solid tumors); active in translational oncology
Chlorotoxin–nanoparticle conjugates	Preclinical/Translational	Glioblastoma, metastatic tumors	Enhanced blood–brain barrier penetration; targeted imaging and therapeutic delivery in animal models	Preclinical; advancing toward first-in-human evaluation
Other venom-derived peptides (e.g., BmK peptides, Bengalin, Neopladines, Maurocalcine)	Preclinical	Leukemia, breast cancer, glioblastoma, colon carcinoma	Demonstrated apoptosis induction, cell cycle arrest, and tumor inhibition in vitro and xenograft models; low toxicity to normal cells	Preclinical only; not yet in clinical trials

## Data Availability

No new data were created or analyzed in this study. Data sharing is not applicable to this article.

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
