# Peer review of "Scorpion Venom as a Source of Cancer Drugs: A Comprehensive Proteomic Analysis and Therapeutic Potential"

_ijms, 2025, doi:10.3390/ijms26209907_

Round 1

Reviewer 1 Report

Comments and Suggestions for Authors

ijms-3868008 – Review

Scorpion Venom as a Source of Cancer Drugs: A Comprehensive Proteomic

Analysis and Therapeutic Potential

 Journal: International Journal of Molecular Sciences.

The manuscript is a scientific review that addresses the therapeutic potential of peptides derived from the venom of scorpions and other animals in cancer treatment, with particular emphasis on gliomas and neuroblastoma. The work presents potentially valuable material (an extensive review of venom peptides and oncological applications). However, several areas require improvement to strengthen the scientific rigor, clarity, and overall impact of the manuscript.

  • The introduction is somewhat lengthy and, in parts, redundant with the “Current Challenges” section (lines 46–68). I recommend condensing it so that the aim of the work becomes explicit early on (e.g., lines 40–44: what does this review add compared to the current state of the art?).
  • The manuscript makes claims consistent with a systematic review and tabulates findings from multiple studies (tables with peptides, preclinical and clinical trials), but it does not include a methods section or search strategy describing: databases consulted (PubMed, Scopus, Web of Science), search terms, cutoff date, inclusion/exclusion criteria, number of articles selected, nor a PRISMA diagram when appropriate. This is essential for reproducibility and bias assessment. The text describes “key steps in venom proteomics” and “This workflow is summarized in Figure 1,” but this refers to an experimental workflow, not the bibliographic strategy of the manuscript. Adding a detailed METHODS section (or, if not a systematic review, clarifying that it is a narrative review and justifying the methodology) is indispensable.
  • Tables need to be standardized and completed. In the peptide tables (e.g., sections listing compounds and mechanisms), several entries contain dashes “–” in key columns (e.g., lack of information on dosage, in vivo vs. in vitro model, number of replicates, analytical method), and formatting is inconsistent across rows. This hinders critical assessment. The tables should be standardized and include at least: species/peptide (sequence) / molecular mass / analytical method/model (cellular/animal) / main endpoints (ICâ‚…â‚€, % tumor reduction) / n / complete reference for each entry.
  • No clear Statistical analysis section is present, nor is there detail on the statistical tests employed in the original studies being summarized. In the preclinical and clinical results sections, percentages, ICâ‚…â‚€ values, and tumor reductions are reported without specifying statistical tests, confidence intervals, or n. This prevents assessment of the robustness of the findings. A statistical section (test used, software, n, criteria for significance) is mandatory.
  • Minor comments: Add DOIs for recent references and correct incomplete references. Standardize nomenclature (e.g., always use ICâ‚…â‚€ consistently). Figure legends should be self-contained (brief methods + summarized results), and the software used to generate the figures should be indicated.

Author Response

We sincerely appreciate the efforts of the reviewer for the insightful feedback and constructive comments, which have allowed us to significantly improve the quality of our paper.
We have carefully considered all the suggestions and have revised the manuscript accordingly. All changes in the text have been highlighted as requested. Below, we provide a point-by-point response to the comments from the reviewer.

COMMENT 1:

The introduction is somewhat lengthy and, in parts, redundant with the “Current Challenges” section (lines 46–68). I recommend condensing it so that the aim of the work becomes explicit early on (e.g., lines 40–44: what does this review add compared to the current state of the art?).

RESPONSE:

We thank the reviewer for this valuable suggestion. We have condensed and restructured the entire introduction, merging the subsections to create a more concise and focused narrative that clearly states the aim of the review early on.

COMMENT 2:

The manuscript makes claims consistent with a systematic review and tabulates findings from multiple studies (tables with peptides, preclinical and clinical trials), but it does not include a methods section or search strategy describing: databases consulted (PubMed, Scopus, Web of Science), search terms, cutoff date, inclusion/exclusion criteria, number of articles selected, nor a PRISMA diagram when appropriate. This is essential for reproducibility and bias assessment. The text describes “key steps in venom proteomics” and “This workflow is summarized in Figure 1,” but this refers to an experimental workflow, not the bibliographic strategy of the manuscript. Adding a detailed METHODS section (or, if not a systematic review, clarifying that it is a narrative review and justifying the methodology) is indispensable.

RESPONSE:

We agree completely with the reviewer on the need for methodological transparency and we thank you for this critical feedback. We have added a new section (Section 2: "Scope and Proteomics-Focused Literature Strategy") that clarifies our bibliographic approach. In this section, we explicitly state that our manuscript is a narrative review. We describe the databases consulted and the keywords used in our literature search. As explained within this new section, a formal systematic methodology like the PRISMA protocol was not employed, as it is not conventional for an expert-driven narrative review. In addition, the reviewer correctly noted that our original Figure 1 depicted an experimental workflow. We have revised this figure to present a more comprehensive overview of the entire venom-to-drug discovery pipeline, which is more aligned with the broad scope of our review manuscript. We are confident that these changes fully address the reviewer's concerns regarding methodological clarity and the manuscript's overall framing.

COMMENT 3:

Tables need to be standardized and completed. In the peptide tables (e.g., sections listing compounds and mechanisms), several entries contain dashes “–” in key columns (e.g., lack of information on dosage, in vivo vs. in vitro model, number of replicates, analytical method), and formatting is inconsistent across rows. This hinders critical assessment. The tables should be standardized and include at least: species/peptide (sequence) / molecular mass / analytical method/model (cellular/animal) / main endpoints (ICâ‚…â‚€, % tumor reduction) / n / complete reference for each entry.

RESPONSE:

We appreciate the reviewer pointing this out. We have thoroughly revised and standardized Table 1 to ensure consistency and completeness. We have also updated the table legend to define abbreviations for missing data (e.g., "NR" for Not Reported), as per standard academic practice.

COMMENT 4:

No clear Statistical analysis section is present, nor is there detail on the statistical tests employed in the original studies being summarized. In the preclinical and clinical results sections, percentages, ICâ‚…â‚€ values, and tumor reductions are reported without specifying statistical tests, confidence intervals, or n. This prevents assessment of the robustness of the findings. A statistical section (test used, software, n, criteria for significance) is mandatory.

RESPONSE:

We thank the reviewer for this comment. As this is a narrative review summarizing findings from numerous published studies, we have not included a separate statistical analysis section, which is conventional for this type of article. However, we have taken care to report the data as accurately as they were presented in the original sources.

COMMENT 5:

Minor comments: Add DOIs for recent references and correct incomplete references. Standardize nomenclature (e.g., always use ICâ‚…â‚€ consistently). Figure legends should be self-contained (brief methods + summarized results), and the software used to generate the figures should be indicated.

RESPONSE:

We thank the reviewer for these important details. We have corrected the reference list, by including DOIs for recent references and incomplete references. We have also standardized the nomenclature for terms like ICâ‚…â‚€ throughout the manuscript, and ensured that all figure legends are self-contained and correctly placed.

Reviewer 2 Report

Comments and Suggestions for Authors

In the review “Scorpion Venom as a Source of Cancer Drugs: A Comprehensive Proteomic Analysis and Therapeutic Potential” Stephanie Santos Suehiro Arcos and coauthors report on the diversity of scorpion toxins and their antitumor potential which can be used for drug design. The study of scorpion venoms is one of the most interesting fields of toxinology due to the wide structural diversity of toxins and is of great practical interest for pharmacology due to their affecting various molecular targets, such as ion channels, receptors, and enzymes. The review includes general aspects about the scorpion venom compositions, descriptions of the structural features and molecular mechanisms of individual venom components exhibiting antitumor activity, and their prospects for use in medicine. Generally, the review is written in a structured good English manner and meets the requirements of the IJMS. However, there are some issues that should be addressed.

  1. In my opinion, the review text is too longer and may be shortened in some sections. Thus, in the Introduction, all subsections should be combined and shortened, while subsection 1.5 may be deleted due to it repeats the information presented in a section 2. Text in Section 4 and 5 contains some repeats of peptides and their activities and should be combined, while new subsections may be separated based on either scorpion toxin targets or mechanism of anticancer activity. In both cases, the information will be well presented.
  2. Authors report on scorpion phospholipase in lines 189-191, then describe disulfide-rich peptides and return to phospholipases again (lines 211-212). In my point of view, subsection 2.2 also should be structured according to the structure groups of proteins and peptides without repeats. Moreover, authors mention the enzyme inhibitors in one sentence (line 188) but do not describe them further. Maybe, this peptide group should be described in more detail.
  3. According to the IJMS template, the Figure caption should be located after the picture. Authors should swap figure and figure caption throughout the text.
  4. Line 801: K_d should be changed to Kd
  5. Line 865: What means “2011” before references?
  6. Line 883: Reference should be in square brackets.
  7. Authors characterize the trial status of TM-601 in Table 2, which discontinued for glioma therapy and redirected to imaging and drug-delivery applications. Maybe this information should be mentioned in the main text too?

Nevertheless, these comments do not diminish the value of the work proposed here. I recommend this paper to be published after minor revision.

Author Response

We sincerely appreciate the efforts of the reviewer for the insightful feedback and constructive comments, which have allowed us to significantly improve the quality of our paper.

We have carefully considered all the suggestions and have revised the manuscript accordingly.

All changes in the text have been highlighted as requested.

Below, we provide a point-by-point response to the comments from the reviewer.

COMMENT 1:

In my opinion, the review text is too longer and may be shortened in some sections. Thus, in the Introduction, all subsections should be combined and shortened, while subsection 1.5 may be deleted due to it repeats the information presented in a section 2. Text in Section 4 and 5 contains some repeats of peptides and their activities and should be combined, while new subsections may be separated based on either scorpion toxin targets or mechanism of anticancer activity. In both cases, the information will be well presented.

RESPONSE:

We are grateful for this suggestion. We have condensed the Introduction and have restructured and combined parts of Sections 4 and 5 to eliminate redundancy and improve the narrative flow, as recommended.

COMMENT 2:

Authors report on scorpion phospholipase in lines 189-191, then describe disulfide-rich peptides and return to phospholipases again (lines 211-212). In my point of view, subsection 2.2 also should be structured according to the structure groups of proteins and peptides without repeats. Moreover, authors mention the enzyme inhibitors in one sentence (line 188) but do not describe them further. Maybe, this peptide group should be described in more detail.

RESPONSE:

We thank the reviewer for identifying this structural issue. We have completely reorganized this section (now Section 3) to group protein and peptide families logically, ensuring that related compounds like phospholipases are discussed together to avoid repetition.

COMMENT 3:

According to the IJMS template, the Figure caption should be located after the picture. Authors should swap figure and figure caption throughout the text.

RESPONSE:

This has been corrected. All figure captions are now placed after the corresponding figure, in accordance with the IJMS template.

COMMENT 4-6 (Minor corrections):

Line 801: K_d should be changed to Kd

Line 865: What means “2011” before references?

Line 883: Reference should be in square brackets.

RESPONSE:

We have corrected all these specific points in the revised manuscript.

COMMENT 7:

Authors characterize the trial status of TM-601 in Table 2, which discontinued for glioma therapy and redirected to imaging and drug-delivery applications. Maybe this information should be mentioned in the main text too?

RESPONSE:

This was an excellent suggestion. We have added a sentence to the main text (Section 5.5.1) to explicitly mention the discontinuation of the TM-601 therapeutic trial, including its clinical trial identifier.

Round 2

Reviewer 1 Report

Comments and Suggestions for Authors

The manuscript has a considerable change, I think it can be accepted as such.